# ELASTIQ: EEG–Language Alignment with Semantic Task Instruction and Querying

## Abstract

Recent advances in electroencephalography (EEG) foundation models, which capture transferable EEG representations, have greatly accelerated the development of brain–computer interfaces (BCI). However, existing approaches still struggle to incorporate language instructions as prior constraints for EEG representation learning, limiting their ability to leverage the semantic knowledge inherent in language to unify different labels and tasks. To address this challenge, we present **ELASTIQ**, a foundation model for **EEG–Language Alignment with Semantic Task Instruction and Querying**. ELASTIQ integrates task-aware semantic guidance to produce structured and linguistically aligned EEG embeddings, thereby enhancing decoding robustness and transferability. In the EEG pretraining stage, we introduce a joint **Spectral–Temporal Reconstruction (STR)** framework that captures the coupled spectral rhythms and temporal dynamics of EEG signals. STR applies randomized spectral perturbation to enhance frequency robustness and uses two complementary temporal objectives to learn both contextual and sequential structure. In the EEG-Language alignment stage, we propose the **Instruction-conditioned Q-Former (IQF)**. This query-based cross-attention transformer injects instruction embeddings into EEG tokens and achieves semantic alignment with textual label embeddings through learnable queries. We evaluate ELASTIQ on 20 datasets spanning motor imagery, emotion recognition, steady-state visual evoked potentials, covert speech, and healthcare tasks. ELASTIQ achieves state-of-the-art performance on 14 of the 20 datasets and obtains the best average results across all five task categories. Importantly, our analyses reveal for the first time that explicit task instructions serve as semantic priors guiding EEG embeddings into coherent and linguistically grounded spaces. The code and pre-trained weights will be released.

## 1 Introduction

Electroencephalography (EEG) provides noninvasive brain dynamics measurement with millisecond-level temporal resolution, making it particularly suitable for applications such as motor imagery (MI) decoding, emotion recognition, and steady-state visual evoked potential (SSVEP) classification. In addition to its high temporal precision, EEG offers the advantages of portability, relatively low cost, and suitability for long-term monitoring. However, EEG suffers from low signal-to-noise ratio, nonstationarity, and large variability across subjects, datasets, and tasks, which has historically limited its generalizability (Edelman et al., 2024). These shortcomings motivate the development of EEG foundation models (EEG-FMs), which aim to leverage large-scale pretraining to learn transferable representations that can

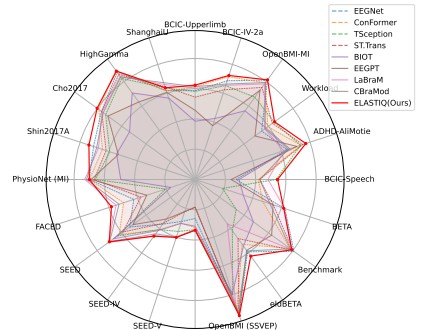

Figure 1: Comparison of ELASTIQ and baseline models on 20 different datasets.

overcome variability and improve downstream task performance. Typically, EEGPT (Wang et al., 2024a) applies transformer-based pretraining to capture temporal dependencies. LaBraM (Jiang et al., 2024c) leverages masked autoencoding on large EEG corpora to learn generalizable embed-

dings. CBraMod (Wang et al., 2024b) focuses on cross-brain modeling to facilitate cross-subject transfer. These EEG-FMs approaches constrained model training with discrete task labels (e.g., 0/1 instead of happy/angry), thereby discarding task-relevant semantic information. This absence of EEG-language coupling constraints may partly result in limited generalization. More recently, Large Language Models (LLMs) have been introduced to further enhance EEG-FMs due to their tremendous success in natural language processing (Touvron et al., 2023) and multimodal understanding (Radford et al., 2021). As a pioneering work, NeuroLM (Jiang et al., 2024b) aligns EEG and language embeddings by training a text-aligned neural tokenizer. Specifically, EEG signals are discretized into tokens and adversarially forced into the same embedding space as text. These EEG tokens are then added to the LLM vocabulary and jointly modeled with text through multi-channel autoregression and instruction tuning. While promising, current EEG-FMs and EEG–language FMs still face two major limitations: First, **missing spectro-temporal interaction**: EEG signals exhibit strong coupling between spectral rhythms and multi-timescale temporal dynamics. These dynamics include causal transitions such as event-related potentials and contextual fluctuations reflecting cognitive or emotional states (Wairagkar et al., 2021; Li et al., 2022). However, existing methods do not jointly model spectral structure together with both contextual and causal temporal dependencies, making it difficult to capture coherent patterns that are essential for reliable EEG representation learning. Second, **insufficient semantic-level alignment**: current EEG-language FMs such as NeuroLM align EEG signals with language through a coarse-grained distribution matching objective. This approach does not incorporate fine-grained semantic constraints and therefore limits the ability of language for guiding EEG representations.

To address these two problems, we propose **ELASTIQ**, a foundation model for EEG–Language Alignment with Semantic Task Instruction and Querying. This approach first introduces a **joint spectral–temporal reconstruction framework** that unifies frequency modeling with both bidirectional and causal temporal learning. By combining global spectral perturbation with complementary temporal masking strategies, the model learns frequency-aware and contextually rich EEG representations, laying a stronger foundation for downstream tasks. To further bridge EEG signals with semantic information, we propose an **Instruction-conditioned Q-Former (IQF)** that matches EEG representations with natural language with semantic alignment. Specifically, EEG embeddings are modulated by the task-level instructions (e.g., "This is an MI task", "Decode emotion from EEG") and label semantics (e.g., "Left", "Happy"), thereby guiding representation learning toward task-relevant dimensions. The modulated EEG features are then refined through a cross-attention mechanism with learnable queries, enabling instruction-driven alignment between EEG and language representations.

Our main contributions are as follows:

- We introduce **ELASTIQ**, a novel EEG–Language Foundation Model designed for EEG decoding across diverse BCI applications. ELASTIQ unifies spectral–temporal modeling with instruction guidance to realize semantic-level EEG-language alignment, thereby enhancing transferability and interpretability across heterogeneous downstream tasks.
- We design two key components to realize this framework: a joint **Spectral–Temporal Reconstruction (STR)** module that jointly captures frequency along with both causal and contextual temporal dynamics, and an **Instruction-conditioned Q-Former (IQF)** that integrates task instructions and label semantics into EEG features through query-based cross-modal alignment.
- We conduct a comprehensive evaluation on 20 EEG datasets spanning motor imagery, emotion, SSVEP, covert speech, and healthcare tasks. ELASTIQ achieves state-of-the-art (SOTA) average performances across all 5 tasks and demonstrates strong generalization across datasets.
- For the first time, we demonstrate that explicit instructions act as semantic priors that restructure EEG feature spaces for better separability, and that stronger text encoders supply richer semantics, leading to faster convergence, higher accuracy, and improved generalization.

## 2 METHOD

In this section, we introduce the design of **ELASTIQ**, our proposed EEG–Language foundation model. ELASTIQ is trained in two stages: an EEG pretraining stage, where a joint spectral–temporal objective encourages frequency-aware and temporally predictive EEG representations, and a multi-task instruction tuning stage, where EEG embeddings are conditioned on task instructions and

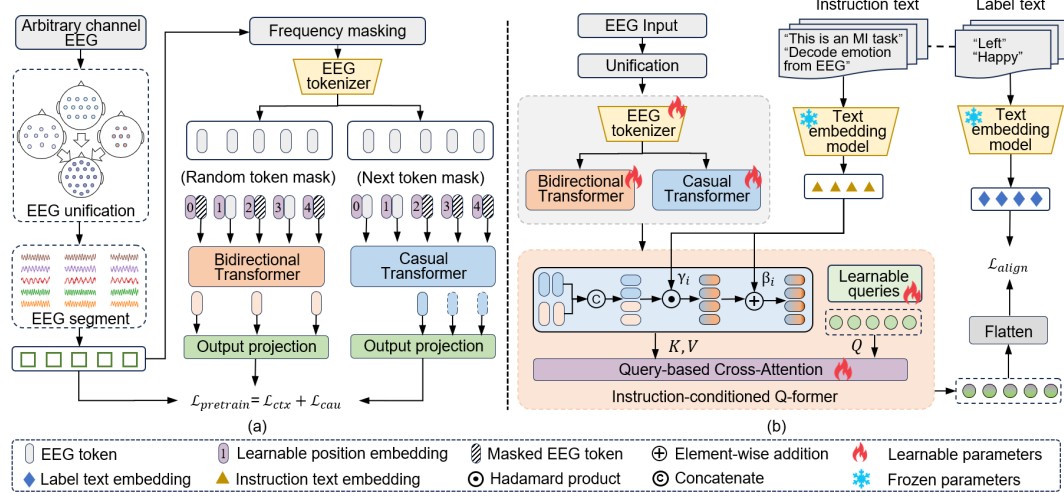

Figure 2: The architecture design of ELASTIQ. **(a)** joint Spectral-Temporal Reconstruction module (STR) for self-supervised EEG pretraining, combining frequency masking, global context modeling, and temporal sequence learning. **(b)** During multi-task instruction tuning, an Instruction-conditioned Q-Former (IQF) aligns EEG signals with language by injecting instruction embeddings and leveraging query-based cross-attention.

aligned with textual targets to improve decoding performance across diverse tasks. The architecture design of ELASTIQ can be found at Figure 2.

## 2.1 EEG PRETRAINING WITH JOINT SPECTRAL-TEMPORAL RECONSTRUCTION

EEG signals combine spectral rhythms with temporal dynamics that manifested both causal transitions and broader contextual fluctuations. To learn representations that capture these spectral–temporal relationships in the pretraining stage, we propose joint spectral-temporal reconstruction module (STR). STR is a two-stream pretraining architecture built upon a shared spectral backbone. The backbone introduces randomized frequency suppression to enforce spectral invariance, while two complementary temporal branches operate in parallel to capture contextual and sequential dependencies. Let $X \in \mathbb{R}^{C \times T}$ denote EEG trial with $C$ means EEG channels and $T$ time points. To manage long recordings and improve training stability, $X$ is segmented into a sequence of non-overlapping windows of fixed length $t$, yielding segments $x_i \in \mathbb{R}^{C \times t}$ for $i = 1, \ldots, \lfloor T/t \rfloor$. Each segment captures synchronized activity across all channels within the temporal window.

**Spectral masking backbone** The spectral masking backbone encourages the encoder to learn frequency-aware features across paradigms. We first suppress a randomly chosen frequency band in each segment before tokenization. For $x_i$, compute its spectrum $X_{f,i} = \text{FFT}(x_i)$ and randomly select a band $[f_{\min}, f_{\max}]$ of width $f_{\text{band}}$ to remove, producing a masked spectrum $\widetilde{X}_{f,i} = \mathcal{M}_{[f_{\min}, f_{\max}]}(X_{f,i})$. Then we conduct an inverse transform to get the perturbed signal via $\widetilde{x}_i = \text{iFFT}(\widetilde{X}_{f,i})$. This encourages invariance to the loss of localized spectral components and complements the dual spectral–temporal objectives. We adopt a lightweight tokenizer consisting of a temporal convolution, a spatial convolution, batch normalization, and pooling:

$$\widetilde{\mathbf{z}}_i = \text{Tokenizer}(\widetilde{x}_i) = Pool\big(BatchNorm(\text{Conv}_S(\text{Conv}_T(\widetilde{x}_i)))\big). \tag{1}$$

The resulting token embeddings $\widetilde{\mathbf{z}}_i$ lie in $\mathbb{R}^{N \times d}$, where $N$ is the number of tokens and $d$ the embedding dimension.

**Dual temporal masking branches** EEG temporal dynamics reflect both causal progression, capturing directed event-related patterns such as onset–sustain–offset transitions in emotional responses, and contextual dependencies that encode longer-range temporal relationships such as emotion states.

To jointly capture these complementary dependencies, STR employs two temporal branches: a bidirectional transformer that reconstructs masked tokens from surrounding context, and a causal transformer that predicts future representations from past context. A bidirectional transformer is trained with a random masking strategy, where a subset of token positions $\mathcal{M}$ is replaced by mask tokens and the model reconstructs the corresponding input token $\widetilde{\mathbf{z}}_i$ from the unmasked context. Each reconstructed token is then mapped back to the input space through a two-layer MLP decoder:

$$g(\widetilde{\mathbf{z}}_i) = W_2 \, \sigma(W_1 \widetilde{\mathbf{z}}_i + b_1) + b_2, \tag{2}$$

where $\sigma(\cdot)$ denotes a non-linear activation, $W_1$ and $W_2$ are learnable weight matrices, and $b_1$ and $b_2$ are the corresponding bias terms. The reconstruction loss is computed against the original input segment:

$$\mathcal{L}_{ctx} = \frac{1}{|\mathcal{M}|} \sum_{i \in \mathcal{M}} \|g(\widetilde{\mathbf{z}}_i) - x_i\|_2^2. \tag{3}$$

A causal transformer is optimized with a future mask, restricting each token at position $i$ to attend only to $\{1, \ldots, i\}$, thus preventing information leakage from the future. This imposes an autoregressive task in which the model predicts the next-token representation $\widetilde{\mathbf{z}}_{i+1}$. The prediction is decoded through the same two-layer MLP, yielding $g(\widetilde{\mathbf{z}}_{i+1})$, and the next-token loss is defined as

$$\mathcal{L}_{cau} = \frac{1}{N-1} \sum_{i=1}^{N-1} \|g(\widetilde{\mathbf{z}}_{i+1}) - x_{i+1}\|_2^2. \tag{4}$$

**Joint spectral–temporal objective**   The overall pretraining objective combines structural and temporal components:

$$\mathcal{L}_{pretrain} = \lambda_{ctx} \mathcal{L}_{ctx} + \lambda_{cau} \mathcal{L}_{cau}, \tag{5}$$

where $\lambda_{\text{ctx}}$ and $\lambda_{\text{cau}}$ are balancing coefficients (set to 1 by default). This design enforces that latent tokens must be decodable through $g(\cdot)$ back into the input domain, ensuring that the learned representations remain both contextually and temporally consistent with the original signals.

## 2.2 EEG-LANGUAGE ALIGNMENT WITH MULTI-TASK INSTRUCTION TUNING

The goal of multi-task instruction tuning is to bridge the gap between EEG and language by learning conditionally aligned representations. To this end, we integrate semantic guidance from textual instructions into EEG embeddings and employ a compact set of latent queries to selectively attend to instruction-relevant neural patterns. The refined neural representation is then projected into a shared semantic space across tasks, where it is aligned with textual label prototypes to enable instruction-grounded decoding. Compared with coarse distribution-level alignment, which merely makes EEG and text statistically indistinguishable, our approach establishes explicit semantic correspondence that allows linguistic priors to directly shape and guide EEG representations.

**Instruction-conditioned EEG-Language Interaction**   Previous EEG–language models such as NeuroLM achieve alignment through adversarial domain matching between neural and textual representations. In contrast, we proposed the Instruction-conditioned Q-Former (IQF). It achieves EEG–language alignment in a constructive and interpretable manner by conditioning neural representations on semantics and guiding cross-modal interaction. Let $\mathbf{m} \in \mathbb{R}^{2N \times d}$ denote the sequence of tokenized EEG embeddings obtained from the pretrained encoder, where $2N$ is the number of concatenated tokens and $d$ is the EEG embedding dimension. Given the instruction text $s_{\text{ins}}$, we obtain its embedding $\mathbf{e}_{\text{ins}} \in \mathbb{R}^k$ using a frozen pretrained language encoder such as BERT (Devlin et al., 2019) or SBERT (Reimers & Gurevych, 2019):

$$\mathbf{e}_{\text{ins}} = f_{\text{text}}(s_{\text{ins}}), \tag{6}$$

and we $\ell_2$-normalize the embedding:

$$\mathbf{e}_{\text{ins}} \leftarrow \frac{\mathbf{e}_{\text{ins}}}{\|\mathbf{e}_{\text{ins}}\|_2}. \tag{7}$$

This normalized vector serves as a high-level semantic prior to guide EEG representations toward the language space.

To fuse this conditioning prior with the EEG embedding space, we employ a Feature-wise Linear Modulation (FiLM) operator (Perez et al., 2017), which parameterizes an affine transformation. Specifically, the modulation parameters $(\boldsymbol{\gamma}, \boldsymbol{\beta})$ are derived from the instruction embedding via a nonlinear projection:

$$(\boldsymbol{\gamma}, \boldsymbol{\beta}) = \tanh\big(\mathbf{W}_{\gamma\beta}\,\mathbf{e}_{\text{ins}} + \mathbf{b}_{\gamma\beta}\big), \tag{8}$$

where $\mathbf{W}_{\gamma\beta} \in \mathbb{R}^{2d \times k}$ and $\mathbf{b}_{\gamma\beta} \in \mathbb{R}^{2d}$. We split $(\boldsymbol{\gamma}, \boldsymbol{\beta})$ into two $d$-dimensional vectors $\boldsymbol{\gamma} \in \mathbb{R}^d$ and $\boldsymbol{\beta} \in \mathbb{R}^d$. For element-wise modulation, these vectors are broadcast from $(d)$ to $(2N \times d)$ so that each latent dimension of every token is modulated by the same semantic prior. The instruction-conditioned representation is then obtained by

$$\tilde{\mathbf{m}} = (1 + \boldsymbol{\gamma}) \odot \mathbf{m} + \boldsymbol{\beta}, \tag{9}$$

where $\odot$ denotes element-wise multiplication. This formulation ensures that $\mathbf{m}$ is shaped by instruction semantics rather than generic alignment. The instruction embedding $\mathbf{e}_{\text{ins}}$ biases the EEG latent space toward task-relevant features, producing representations that are both aligned with textual targets $\mathbf{e}_{\text{tgt}}$ and regularized on an instruction-informed manifold for improved semantic fidelity and generalization.

To extract instruction-relevant neural features from high-dimensional modulated EEG embeddings, we introduce a set of $N_q$ learnable query vectors $\mathbf{Q}_0 \in \mathbb{R}^{N_q \times d}$, which function as compact latent probes. Rather than directly inheriting the full complexity of the EEG embedding space, these queries serve as bottlenecks through which information must be filtered. The proposed IQF employs cross-attention to couple $\mathbf{Q}_0$ with the instruction-modulated EEG embeddings $\tilde{\mathbf{m}}$, yielding

$$\mathbf{Q} = \text{QFormer}(\mathbf{Q}_0, \tilde{\mathbf{m}}) = \text{softmax}\left( \frac{\mathbf{Q}_0 W_Q\,(\tilde{\mathbf{m}}W_K)^\top}{\sqrt{d}} \right) \tilde{\mathbf{m}}W_V, \tag{10}$$

where $W_Q, W_K, W_V$ denote the query, key, and value projections, and $d$ is the key dimension. Through this operation, each query selectively attends to instruction-relevant EEG patterns encoded in $\tilde{\mathbf{m}}$ effectively performing instruction-guided feature extraction. Conceptually, this operation projects EEG embeddings onto a lower-dimensional query subspace regularized by the instruction prior. The learnable queries act as semantic filters, retaining task-relevant features while suppressing irrelevant variance. This constrained information flow can be viewed as conditional information maximization, yielding embeddings that are semantically consistent and generalizable. The resulting query outputs $\mathbf{Q}$ are aggregated and projected through a lightweight MLP to produce the instruction-aligned EEG representation $\hat{\mathbf{h}}$:

$$\hat{\mathbf{h}} = \frac{\mathbf{h}}{\|\mathbf{h}\|_2}, \quad \mathbf{h} = f_{\text{MLP}}(\text{vec}(\mathbf{Q})) \tag{11}$$

where $\|\cdot\|_2$ denotes the $\ell_2$ (Euclidean) norm.

paragraphTextual Prototype-based Semantic Alignment Given the ground-truth label $y \in \mathcal{C}$, we obtain its textual prototype embedding $\mathbf{e}_{\text{tgt}} \in \mathbb{R}^k$ by encoding the corresponding class name with the same frozen language model used for instructions:

$$\mathbf{e}_{\text{tgt}} = f_{\text{text}}(s_{\text{tgt}}), \tag{12}$$

Using the same encoder for both instructions and labels ensures that they are represented in a shared semantic space. To align $\mathbf{h}$ with its semantic prototype, we minimize a cosine similarity loss between them:

$$\mathcal{L}_{\text{align}} = \frac{1}{|\mathcal{C}|} \sum_{c \in \mathcal{C}} \left( 1 - \cos(\hat{\mathbf{h}}, \mathbf{e}_{tgt}^c) \right) \mathbb{I}[y = c], \tag{13}$$

where $\mathbb{I}[y = c]$ is the indicator function for the ground-truth class. This objective enforces $\hat{\mathbf{h}}$ to be maximally aligned with its corresponding prototype $\mathbf{e}_{\text{tgt}}$ while remaining orthogonal to irrelevant ones. As a result, EEG embeddings and language prototypes cohabit a shared latent space $\mathcal{Z}$, where distances reflect cross-modal semantic consistency and class-level discriminability.

## 2.3 INFERENCE PROCEDURE

To evaluate the proposed ELASTIQ, we consider two inference regimes: (1) **Task-specific fine-tuning**: following multi-task instruction tuning, the model is further fine-tuned on the training split of each target dataset to adapt to its domain-specific distribution, while maintaining the same instruction–label formulation. This process does not introduce any additional classification head but refines the shared EEG–language parameters to better align with the target domain. (2) **Direct inference**: the multi-task instruction-tuned model is kept frozen and directly evaluated on the test set of each downstream dataset. Given the corresponding textual instructions and target label embeddings $t_y$, predictions are obtained through cosine similarity between $\tilde{z}_{\text{EEG}}$ and $t_y$. In both regimes, the final prediction is computed as

$$\hat{y} = \arg\max_y \cos(\tilde{z}_{\text{EEG}}, t_y),$$

where $t_y$ is the embedding of the textual label (e.g., "left hand", "right hand").

Notably, the language model used in ELASTIQ is fully frozen, and all instruction and label embeddings are pre-computed offline, so it does not participate in either training or inference. Consequently, ELASTIQ introduces no extra computational overhead from the text encoder, and the resulting model remains lightweight.

# 3 EXPERIMENTS

## 3.1 DATASET

**Pretraining dataset**  We use 9 datasets, namely Stieger2021 (Stieger et al., 2021), SEED-FRA (Liu et al., 2022b), SEED-GER (Liu et al., 2022b), SEED-SD (Li et al., 2025), SEED-Neg, ChineseEEG (Mou et al., 2024), Chisco (Zhang et al., 2024), LargeSpanish(Valle et al., 2024), ThinkOutLoud (Nieto et al., 2022) as the pretraining datasets. The total duration of these datasets is around 1153 hours. More details about the pretraining datasets can be found in Appendix M.

**Downstream Dataset**  We systematically evaluate our ELASTIQ on the five different BCI tasks with 20 datasets in total. **Motor Imagery**: OpenBMI-MI (Lee et al., 2019), BCIC-IV-2a (Tangermann et al., 2012), BCIC-Upperlimb (Jeong et al., 2022), SHU-MI (Ma et al., 2022), HighGamma (Schirrmeister et al., 2017), Cho2017 (Cho et al., 2017), Shin2017A (Shin et al., 2016), PhysioNet-MI (Schalk et al., 2004). **Emotion**: SEED (Duan et al., 2013), SEED-IV (Zheng et al., 2018), SEED-V (Liu et al., 2021), SEED-VII (Jiang et al., 2024a), FACED (Chen et al., 2023). **SSVEP**: OpenBMI-SSVEP (Lee et al., 2019), BETA (Liu et al., 2020), eldBETA (Liu et al., 2022a), Benchmark (Wang et al., 2016). **Covert speech**: BCIC2020-3 (Jeong et al., 2022). **Healthcare**: ADHD-AliMotie (Nasrabadi et al., 2020), Mental Workload (Zyma et al., 2019). More details about the downstream datasets can be found in Appendix O.

## 3.2 EXPERIMENTAL SETUP

**Baselines & Metrics**  In this paper, we selected both the state-of-the-art traditional models and the EEG-FMs as baselines. For the traditional models, we selected EEGNet (Lawhern et al., 2018), TSception (Ding et al., 2022), ST-Transformer (Song et al., 2021) and Conformer (Song et al., 2022). For the EEG foundation model, we selected BIOT (Yang et al., 2023), EEGPT (Wang et al., 2024a), LaBraM (Jiang et al., 2024c), and CBraMod (Wang et al., 2024b). Implementation about the baselines can be found in Appendix N. To provide a reliable evaluation across imbalanced datasets, we adopted **balanced accuracy** and **Cohen's Kappa** as performance metrics. Balanced accuracy accounts for class imbalance by averaging recall across classes, while Cohen's Kappa measures the agreement between predicted and true labels beyond chance level, providing a more robust assessment of model performance.

**EEG Preprocessing and Unification**  EEG recordings from different studies typically use diverse electrode montages. ELASTIQ performs channel unification by interpolating all signals onto the standardized 10–10 electrode layout, with 65 channels in Appendix S. For datasets recorded with fewer than 65 channels, we perform spatial interpolation to enforce a consistent topological structure across inputs. Then, signals are downsampled to 200 Hz. The MI datasets are band-pass filtered to

Table 1: Results across datasets with traditional and foundation EEG models. To facilitate comparison, the best three results per dataset are highlighted, where darker shading corresponds to higher performance.

| Dataset | Metrics | Traditional models | | | | EEG Foundation models | | | | |
|---|---|---|---|---|---|---|---|---|---|---|
| | | EEGNet | Conformer | TScep. | STTran. | BIOT | EEGPT | LaBraM | CBraMod | ELASTIQ |
| BCIC-IV-2a | B-Acc | 0.6369 | 0.6347 | 0.6252 | 0.5816 | 0.5139 | 0.5279 | 0.6234 | 0.6139 | 0.6381 |
| | B-Acc | 0.4685 | 0.4663 | 0.4543 | 0.4334 | 0.3898 | 0.4617 | 0.4599 | 0.4385 | 0.4692 |
| OpenBMI-MI | B-Acc | 0.8170 | 0.8274 | 0.6665 | 0.7514 | 0.5613 | 0.7323 | 0.7874 | 0.7895 | 0.8134 |
| | B-Acc | 0.6259 | 0.6387 | 0.3425 | 0.5046 | 0.1225 | 0.4624 | 0.5765 | 0.5828 | 0.6127 |
| BCIC-Upperlimb | B-Acc | 0.5281 | 0.5473 | 0.5199 | 0.5284 | 0.4595 | 0.5231 | 0.5414 | 0.5454 | 0.5483 |
| | B-Acc | 0.2962 | 0.3120 | 0.2792 | 0.2972 | 0.1843 | 0.2031 | 0.3108 | 0.3145 | 0.3187 |
| SHU-MI | B-Acc | 0.5665 | 0.5780 | 0.5564 | 0.5714 | 0.5587 | 0.5228 | 0.6338 | 0.6403 | 0.6134 |
| | B-Acc | 0.1976 | 0.1591 | 0.2047 | 0.2191 | 0.2151 | 0.1657 | 0.2338 | 0.2384 | 0.2267 |
| HighGamma | B-Acc | 0.7856 | 0.7637 | 0.6851 | 0.7019 | 0.5824 | 0.6516 | 0.6939 | 0.7684 | 0.7982 |
| | B-Acc | 0.5838 | 0.5627 | 0.5129 | 0.5162 | 0.4855 | 0.5152 | 0.5200 | 0.5671 | 0.5965 |
| Cho2017 | B-Acc | 0.7644 | 0.7838 | 0.7339 | 0.7612 | 0.5413 | 0.7197 | 0.7614 | 0.7439 | 0.7908 |
| | B-Acc | 0.5310 | 0.5719 | 0.4640 | 0.5260 | 0.0729 | 0.4305 | 0.5221 | 0.4974 | 0.5816 |
| Shin2017A | B-Acc | 0.7054 | 0.6464 | 0.5992 | 0.6168 | 0.5422 | 0.5389 | 0.6744 | 0.6861 | 0.7256 |
| | B-Acc | 0.4521 | 0.2901 | 0.1891 | 0.2371 | 0.0909 | 0.0688 | 0.3607 | 0.3689 | 0.4811 |
| PhysioNet-MI | B-Acc | 0.6953 | 0.6951 | 0.6649 | 0.6706 | 0.4874 | 0.6820 | 0.7246 | 0.7219 | 0.6992 |
| | B-Acc | 0.3980 | 0.3952 | 0.3216 | 0.3370 | 0.0162 | 0.3730 | 0.4487 | 0.4386 | 0.3983 |
| FACED | B-Acc | 0.4271 | 0.4943 | 0.2056 | 0.3791 | 0.1711 | 0.3346 | 0.5457 | 0.5787 | 0.5819 |
| | B-Acc | 0.3512 | 0.4263 | 0.1088 | 0.2999 | 0.0647 | 0.2486 | 0.4809 | 0.4941 | 0.5243 |
| SEED | B-Acc | 0.5337 | 0.6254 | 0.6369 | 0.5882 | 0.6674 | 0.5054 | 0.7083 | 0.7102 | 0.7011 |
| | B-Acc | 0.3143 | 0.4342 | 0.4604 | 0.3873 | 0.5034 | 0.2659 | 0.5613 | 0.5868 | 0.5543 |
| SEED-IV | B-Acc | 0.3651 | 0.4094 | 0.4063 | 0.3616 | 0.4141 | 0.3202 | 0.4415 | 0.4605 | 0.4630 |
| | B-Acc | 0.1578 | 0.2121 | 0.1876 | 0.1415 | 0.1937 | 0.0861 | 0.2560 | 0.2728 | 0.2754 |
| SEED-V | B-Acc | 0.2932 | 0.3060 | 0.3637 | 0.2244 | 0.3045 | 0.2253 | 0.4010 | 0.4029 | 0.4126 |
| | B-Acc | 0.1136 | 0.1335 | 0.1984 | 0.0344 | 0.1306 | 0.0335 | 0.2563 | 0.2570 | 0.2575 |
| SEED-VII | B-Acc | 0.2587 | 0.3209 | 0.3300 | 0.1867 | 0.3096 | 0.1809 | 0.3244 | 0.3311 | 0.3356 |
| | B-Acc | 0.1413 | 0.2081 | 0.2198 | 0.0516 | 0.1939 | 0.0476 | 0.2159 | 0.2233 | 0.2275 |
| OpenBMI-SSVEP | B-Acc | 0.9430 | 0.8981 | 0.8026 | 0.9398 | 0.7936 | 0.9301 | 0.8698 | 0.9205 | 0.9462 |
| | B-Acc | 0.9253 | 0.8634 | 0.7375 | 0.9195 | 0.7258 | 0.9098 | 0.8269 | 0.8924 | 0.9283 |
| eldBETA | B-Acc | 0.5922 | 0.5329 | 0.4205 | 0.4848 | 0.5476 | 0.5821 | 0.3653 | 0.5817 | 0.6262 |
| | B-Acc | 0.5410 | 0.4736 | 0.3476 | 0.4199 | 0.4901 | 0.5292 | 0.2848 | 0.5299 | 0.5801 |
| Benchmark | B-Acc | 0.8018 | 0.7691 | 0.3329 | 0.7902 | 0.4771 | 0.6200 | 0.7692 | 0.7436 | 0.7702 |
| | B-Acc | 0.7969 | 0.7635 | 0.3161 | 0.7850 | 0.4637 | 0.6105 | 0.7462 | 0.7379 | 0.7802 |
| BETA | B-Acc | 0.6291 | 0.4710 | 0.1911 | 0.5419 | 0.3358 | 0.5453 | 0.5572 | 0.5239 | 0.5483 |
| | B-Acc | 0.6193 | 0.4574 | 0.1704 | 0.5303 | 0.3187 | 0.5265 | 0.5395 | 0.5116 | 0.5392 |
| BCIC-Speech | B-Acc | 0.2699 | 0.4170 | 0.5314 | 0.4266 | 0.2920 | 0.2364 | 0.4819 | 0.4280 | 0.5453 |
| | B-Acc | 0.0880 | 0.2722 | 0.4164 | 0.2832 | 0.1138 | 0.0478 | 0.3863 | 0.2860 | 0.4317 |
| ADHD-AliMotie | B-Acc | 0.6349 | 0.7316 | 0.7333 | 0.7515 | 0.6516 | 0.7124 | 0.6194 | 0.6434 | 0.7699 |
| | B-Acc | 0.2789 | 0.4681 | 0.4752 | 0.5047 | 0.3155 | 0.4217 | 0.2285 | 0.3121 | 0.5282 |
| Mental Workload | B-Acc | 0.5480 | 0.5984 | 0.6480 | 0.6335 | 0.5857 | 0.4925 | 0.5650 | 0.5746 | 0.6493 |
| | B-Acc | 0.0982 | 0.1801 | 0.3034 | 0.2683 | 0.1480 | 0.0223 | 0.1643 | 0.1695 | 0.3134 |
| Average | B-Acc | 0.5873 | 0.6008 | 0.5278 | 0.5742 | 0.4886 | 0.5292 | 0.6035 | 0.6208 | 0.6494 |
| | Kappa | 0.4118 | 0.4246 | 0.3307 | 0.3886 | 0.2612 | 0.3303 | 0.4308 | 0.4506 | 0.4912 |

0.3–40 Hz, while all other datasets are filtered to 0.3–70 Hz. For segmentation, we distinguish between pre-segmented and continuous datasets. For pre-segmented datasets such as MI, SSVEP, and Covert Speech, we directly use the original trial-based divisions provided by the dataset. For continuous datasets, we follow dataset-specific conventions: FACED recordings are divided into 10-second windows, the SEED series datasets are split into 4-second segments, and the Workload dataset is segmented into 5-second windows. More details are provided in Appendix Q.

**Implementation Details**    Pre-training and instruction tuning are both conducted in an end-to-end manner. Detailed model hyperparameters and training parameters are provided in Appendix K. Our model is trained using a 4xH100 cluster using PyTorch with parameters around 26.42 M.

### 3.3 EXPERIMENTAL RESULTS

**Task-specific finetune**    In the task-specific finetuning setting, ELASTIQ is first trained with multi-task instruction tuning on the combined training split of all datasets. It is then further finetuned using only the training set of each target downstream dataset. This setup ensures a fair comparison against

Table 2: Direct inference performance under different instruction conditions. Results are reported in B-Acc/Kappa. Best results are highlighted in bold.

| Dataset | # Class | No-Instruction | Task-only Instruction | Task & Target Instruction |
|---|---|---|---|---|
| OpenBMI-MI | 2 | 0.5813 / 0.1625 | 0.6742 / 0.3483 | **0.7213 / 0.4654** |
| BCIC-IV-2a | 2 | 0.3516 / 0.1354 | 0.3898 / 0.1863 | **0.4262 / 0.2350** |
| SHU-MI | 2 | 0.5051 / 0.0102 | 0.5170 / 0.0340 | **0.5288 / 0.0574** |
| HighGamma | 2 | 0.5534 / 0.1069 | 0.5373 / 0.0745 | **0.5858 / 0.1716** |
| Cho2017 | 2 | 0.6383 / 0.2767 | 0.7372 / 0.4744 | **0.7337 / 0.4756** |
| Shin2017A | 2 | 0.5417 / 0.0833 | 0.6556 / 0.3111 | **0.6611 / 0.3222** |
| PhysioNet-MI | 2 | 0.5319 / 0.0639 | 0.5485 / 0.0969 | **0.5576 / 0.1152** |
| SEED | 3 | 0.4452 / 0.1735 | 0.4856 / 0.2342 | **0.5659 / 0.3574** |
| SEED-IV | 4 | 0.3172 / 0.1032 | 0.3626 / 0.1724 | **0.3822 / 0.2253** |
| SEED-V | 5 | 0.2796 / 0.1030 | 0.3054 / 0.1343 | **0.3401 / 0.1805** |
| SEED-VII | 7 | 0.1976 / 0.0627 | 0.2251 / 0.0963 | **0.2525 / 0.1282** |
| OpenBMI-SSVEP | 4 | 0.8094 / 0.7458 | 0.8488 / 0.7983 | **0.8506 / 0.8008** |
| Average | - | 0.4794 / 0.1689 | 0.5239 / 0.2468 | **0.5531 / 0.2946** |

baselines including EEGNet, Conformer, TSception, ST-Transformer, BIOT, EEGPT, LaBraM, and CBraMod. All averaged accuracies are reported over three random seeds in Table 1. **Motor Imagery:** Results show that our proposed method achieves the state-of-the-art performance on five MI datasets. Specifically, our method achieves the best performance improvement in BCIC-IV-2a (63.81%), BCIC-Upperlimb (54.83%), HighGamma (79.82%), Cho2017 (79.08%), and Shin2017A (72.56%). **Emotion Recognition:** Results show that our proposed method achieves the state-of-the-art performance on four emotion datasets. Specifically, our method provides the best performance improvement in FACED (58.19%), SEED-IV (46.3%), SEED-V (41.26%), and SEED-VII (33.56%). **SSVEP:** Results show that ELASTIQ achieves the state-of-the-art performance on two SSVEP datasets. Specifically, our method provides great performance improvement in OpenBMI-SSVEP (94.62%), eldBETA (62.62%). **Covert Speech:** Results show that our proposed method achieves the state-of-the-art performance (54.53%) on this covert speech dataset. **Healthcare:** Results show that ELASTIQ achieves the state-of-the-art performance on two datasets (76.99% on ADHD-AliMotie, 64.93% on Mental Workload). In terms of overall performance, ELASTIQ attains an average macro-accuracy of 66.78% and an average Kappa of 53.91%, substantially surpassing the other baselines, thus demonstrating superior generalization across diverse EEG decoding scenarios. Due to space constraints, the results with standard deviations are provided in Table 6 and Table 7 of the appendix.

**Direct inference**     In the direct inference setting, ELASTIQ is trained on the combined training split of all datasets during the multi-task instruction tuning stage and is then directly evaluated on each downstream dataset without any additional finetuning. Since the instruction condition will influence direct inference performance, we report results under three instruction levels: No-instruction: input *Default* or *None*; Task-only instruction: provide an instruction specifying the EEG task; Task & Target Instruction: provide both the task type and target classes (e.g., This is an MI task; decode *Left* vs. *Right*). The quantitative results in Table 2 indicate under the instruction with task and target information, ELASTIQ achieves average accuracy of 55.31%. We can also notice that compared to the "No Instruction" condition, providing task-level instructions yields an average gain of 1.77% in balanced accuracy, while the most detailed condition (task & targets) further improves performance to 2.12%, respectively.

## 3.4 Analysis of Instruction-Guided EEG Representations: Qualitative and Quantitative Evidence

To more comprehensively understand how instructions shape the latent EEG representation space, we present both qualitative visualizations and quantitative metrics. Figure 4 shows a UMAP (McInnes et al., 2018) projection of the learned embeddings, where kernel density estimation (KDE) highlights the concentration of samples. Class labels are represented by fixed textual prototypes that are projected through the same model and UMAP mapping. Without instructions, the embeddings exhibit weak structure, with substantial overlap between classes. When instructions are provided, however, the feature space becomes more organized: class clusters become clearly separated and better aligned with their corresponding semantic prototypes. Beyond visualization,

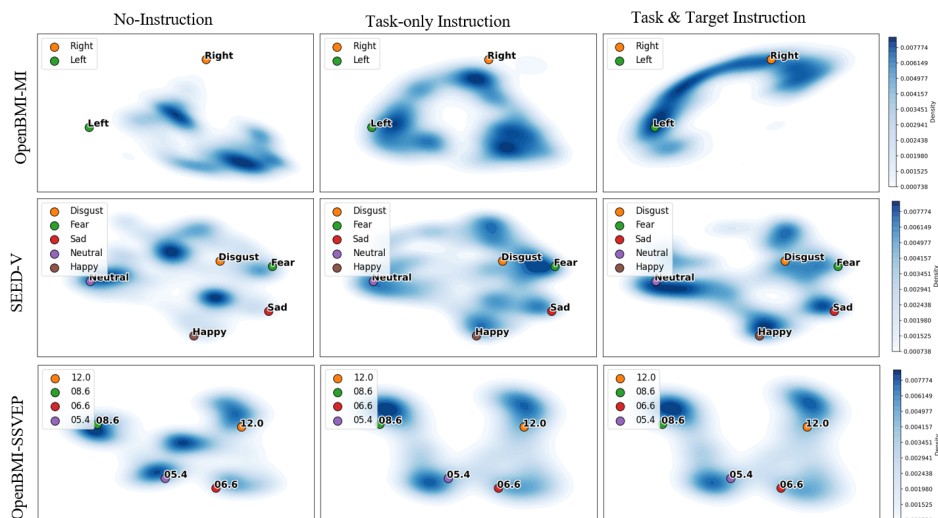

Figure 3: KDE visualization of EEG embeddings under different instructions.

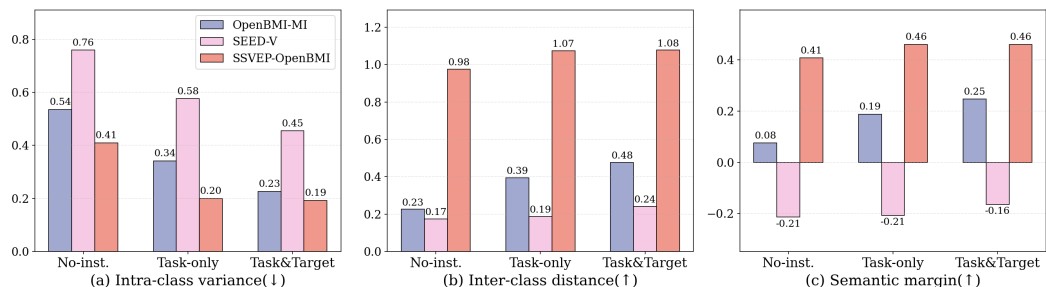

Figure 4: Quantitative results on the effect of language-guided instructions on EEG feature distributions.

we additionally quantify the effects of instruction conditioning using three embedding-level metrics. Specifically, we evaluate: (i) Intra-class variance (↓ better), which measures the compactness of samples within each class; (ii) Inter-class distance (↑ better), which quantifies the separation between class centers; (iii) Semantic margin (↑ better), defined as the difference between true-class and wrong-class similarities. Details of these metrics are provided in Appendix P. Across all metrics, instruction-conditioned models consistently exhibit lower intra-class variance, larger inter-class distances, and substantially higher semantic margin. Both the qualitative and quantitative results prove that natural-language instructions act as explicit semantic constraints that reshape the EEG embedding space toward a more discriminative and semantically meaningful organization.

## 3.5 Effect of Text Embeddings on ELASTIQ

To evaluate the contribution of text embeddings, we train a baseline model that performs multi-task learning using the cross-entropy loss. This experiment allows us to disentangle whether the performance gains arise primarily from the multi-task learning setup or from the incorporation of language semantics. Comparisons are conducted under both direct inference and task-specific finetune settings. Results in Table 3 show that across datasets and settings, ELASTIQ consistently outperforms the baseline without language embedding. We further evaluated several modern text encoders—all-mpnet-base-v2 (Reimers & Gurevych, 2019), bert-base-uncased (Devlin et al., 2019), E5-large-v2 (Wang et al., 2023), Qwen3-embedding-0.6B (Bai et al., 2024), and Voyage-3-large (VoyageAI, 2024)—as frozen embedding backbones for ELASTIQ. Among them, all-mpnet-base-v2 achieves the most stable convergence and highest decoding accuracy. This aligns with its design: MPNet-based SBERT models are optimized for short, sentence-level semantic similarity, which matches the nature of ELASTIQ's task instructions and label phrases (e.g., "left hand", "happy", "12 Hz"). In

Table 3: Comparison of ELASTIQ with different text embeddings under task-specific finetune and direct inference settings. Best results are highlighted in bold.

| Method | OpenBMI-MI | | SEED | |
|---|---|---|---|---|
| | Task-specific finetune | Direct inference | Task-specific finetune | Direct inference |
| Without text embedding | 0.7912 / 0.5987 | - / - | 0.6328 / 0.4581 | - / - |
| ELASTIQ (bert-base-uncased) | 0.8077 / 0.6154 | 0.5156 / 0.0312 | 0.6448 / 0.4729 | 0.4328 / 0.1594 |
| ELASTIQ (all-mpnet-base-v2) | **0.8144 / 0.6287** | **0.7340 /0.4679** | **0.7011 / 0.5543** | 0.5646 / 0.3553 |
| ELASTIQ (voyage-3-large) | 0.8059 / 0.6198 | 0.7124 / 0.4521 | 0.6897 / 0.5402 | 0.5518 / 0.3479 |
| ELASTIQ (e5-large-v2) | 0.7993 / 0.6104 | 0.6831 / 0.4217 | 0.6742 / 0.5235 | 0.5491 / 0.3408 |
| ELASTIQ (qwen3-embedding-0.6b) | 0.7938 / 0.6049 | 0.6975 / 0.4482 | 0.6613 / 0.5126 | **0.5687 / 0.3444** |

contrast, E5-large-v2, Qwen3-embedding, and Voyage-3-large are geared toward retrieval or multilingual embedding, resulting in embedding geometries less suited for the fine-grained semantic alignment required in our setting.

### 3.6 Effect of Joint Spectral-Temporal Reconstruction Module

To evaluate the contribution of each strategy during STR pretraining, we performed an ablation study in which individual masking components were selectively removed. Results in Table 4 indicate that across OpenBMI-MI, BCI-IV-2a, and SEED, retaining all three masking strategies yields the best B-Acc/Kappa. These results indicate that all three components contribute to the model's performance.

Table 4: Ablation study of masking strategies. Best results are highlighted in bold.

| Frequency Mask | Random Mask | Causal Mask | OpenBMI-MI | BCI-IV-2a | SEED | SEED-IV |
|---|---|---|---|---|---|---|
| ✓ | | | 0.8078 / 0.6195 | 0.7093 / 0.6181 | 0.6882 / 0.5487 | **0.4689 / 0.2812** |
| ✓ | ✓ | | 0.7896 / 0.6017 | 0.7015 / 0.6094 | 0.6771 / 0.5298 | 0.4462 / 0.2591 |
| | ✓ | ✓ | 0.8033 / 0.6152 | 0.7168 / 0.6265 | 0.6957 / 0.5499 | 0.4598 / 0.2721 |
| ✓ | ✓ | ✓ | **0.8144 / 0.6287** | **0.7234 / 0.6311** | **0.7011 / 0.5543** | 0.4630 / 0.2754 |

### 3.7 Comparison with NeuroLM

NeuroLM is among the earliest EEG–language alignment models, so we compare ELASTIQ to NeuroLM (Jiang et al., 2024b) under direct inference setting. In Table 5, ELASTIQ surpasses NeuroLM on SEED but underperforms on Workload. ELASTIQ is different from NeuroLM because it avoids adversarial distribution-level alignment and instead uses an instruction-conditioned Q-Former to align EEG signals with textual prototypes, enabling fine-grained and instruction-controllable semantic mapping. In terms of model size, whereas NeuroLM performs generation-based prediction with large language models (254M–1696M), ELASTIQ uses similarity-based classification with frozen text embeddings, reducing the model to 26.4M.

Table 5: Direct inference results comparison. Best results are highlighted in bold.

| Methods | #Params | SEED | | Workload | |
|---|---|---|---|---|---|
| | | Balanced Acc. | Cohen's Kappa | Balanced Acc. | Cohen's Kappa |
| NeuroLM | 254 M | 0.5554±0.0075 | 0.3393±0.0117 | **0.6172±0.0113** | **0.5824±0.0080** |
| ELASTIQ (ours) | 26.4 M | **0.5659±0.0042** | **0.3574±0.0073** | 0.5383±0.0065 | 0.0766±0.0084 |

## 4 Conclusion

We introduced ELASTIQ, a foundation model for EEG–Language Alignment with Semantic Task Instruction and Querying. By combining a joint Spectral-Temporal Reconstruction module and an Instruction-conditioned Q-Former, ELASTIQ learns language-guided EEG representations that transfer effectively across tasks. Extensive evaluations on 20 datasets covering MI, emotion recognition, SSVEP, covert speech, and healthcare applications show that ELASTIQ achieves, on average, state-of-the-art (SOTA) performance, highlighting the value of instruction-informed alignment for generalizable EEG decoding. More importantly, our work positions natural language as both an interpretable anchor and a transferable supervision signal, providing valuable guidance for the development of future EEG foundation models and BCI systems.

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

## A  RELATED WORK

**Self-supervised Pretaining.**  Self-supervised pretraining has emerged as a powerful paradigm in representation learning, reducing the reliance on large amounts of annotated data while leveraging abundant unlabeled signals. Self-supervised methods design pretext tasks that encourage models to learn meaningful feature representations from the inherent structure of data. Early successes in natural language processing, such as BERT (Devlin et al., 2019) and GPT series(Radford et al., 2019), demonstrated that masked language modeling and next-word prediction can yield representations transferable to diverse downstream tasks. Similarly, in computer vision, contrastive learning approaches like SimCLR(Chen et al., 2020), MoCo(He et al., 2020) and masked image modeling like MAE(He et al., 2022) showed that pretraining on large-scale unlabeled images leads to robust and generalizable visual features.

**EEG Foundation model.**  The concept of foundation models has recently expanded into the EEG domain, aiming to build large-scale pre-trained backbones that generalize across datasets, tasks, and clinical conditions. Several pioneering efforts have been proposed. BIOT (Yang et al., 2023) explored scalable transformer-based architectures for biomedical signals, positioning EEG as a central modality. EEGPT (Wang et al., 2024a), inspired by advances in language modeling, leveraged transformer pretraining strategies such as masked prediction and contrastive learning to enhance generalization across heterogeneous EEG datasets. LaBraM (Jiang et al., 2024c) introduced a large-brain-model framework, emphasizing cross-dataset pretraining to capture universal EEG representations. CBraMod (Wang et al., 2024b) extended this idea by focusing on cross-brain modularity, enabling adaptation across diverse cognitive and motor tasks. Beyond EEG-specific approaches, NeuroLM(Jiang et al., 2024b) proposed a broader neural language model for neuroscience data, while PhysioOmni(Jiang et al., 2025) further expanded the scope to multi-physiological modalities, integrating EEG with signals such as ECG and EMG to learn cross-modal representations. Collectively, these efforts highlight the emerging trajectory of EEG-FMs: moving from task-specific networks toward unified, pre-trained architectures capable of powering downstream applications with minimal fine-tuning, and paving the way for general-purpose brain decoding systems.

## B  MEAN AND STANDARD DEVIATION FROM DIFFERENT SEEDS

Since our model is evaluated across 20 datasets, the main paper cannot accommodate the full set of values due to space constraints. We summarized the complete statistics with standard deviations in Table 6 and Table 7 for reference.

## C  CROSS-VALIDATION COMPARISON WITH BEST BASELINE

To supplement the main cross-subject results in Table 1, which follow the same data split protocol as prior work, we further conduct a cross-validation evaluation to provide a more comprehensive comparison. Specifically, we compare ELASTIQ with the strongest baseline, CBraMod, under a standard five-fold cross-validation setting. For datasets without predefined folds, we randomly divide subjects into five approximately equal subsets and ensure that no subject appears in both training and validation splits. For each fold, both ELASTIQ and CBraMod are trained on four folds and evaluated on the remaining fold, and we report the mean and standard deviation across folds. We selected three datasets representing three distinct EEG tasks: OpenBMI-MI (motor imagery), FACED (emotion), and OpenBMI-SSVEP (SSVEP detection). As shown in Table 8, ELASTIQ consistently surpasses CBraMod across all datasets. On OpenBMI-MI, ELASTIQ achieves $0.8270 \pm 0.016$, significantly higher than CBraMod's $0.7840 \pm 0.0272$ (Wilcoxon $p = 0.0072$, $p < 0.01$). On FACED, ELASTIQ attains $0.5680 \pm 0.029$, outperforming CBraMod's $0.5510 \pm 0.0138$ with statistical significance ($p = 0.0435$, $p < 0.05$). For the OpenBMI-SSVEP dataset, ELASTIQ reaches $0.9250 \pm 0.013$, again exceeding CBraMod's $0.9124 \pm 0.0253$ ($p = 0.0232$, $p < 0.05$). These results show that ELASTIQ not only improves cross-subject generalization but also maintains robustness across different subject partitions. The statistically significant gains across multiple datasets confirm that ELASTIQ provides consistent and reliable improvements over the strongest existing EEG foundation model.

Table 6: Performance comparison on datasets (Balanced Accuracy and Cohen's Kappa).

| Methods | BCIC-IV-2a | | OpenBMI-MI | | BCIC-Upperlimb | |
| --- | --- | --- | --- | --- | --- | --- |
| | B.Acc | Kappa | B.Acc | Kappa | B.Acc | Kappa |
| EEGNet | 0.6369 ± 0.0097 | 0.4685 ± 0.0130 | 0.8170 ± 0.0032 | 0.6259 ± 0.0114 | 0.5281 ± 0.0046 | 0.2962 ± 0.0062 |
| Conformer | 0.6347 ± 0.0118 | 0.4663 ± 0.0157 | 0.8274 ± 0.0027 | 0.6387 ± 0.0098 | 0.5473 ± 0.0039 | 0.3120 ± 0.0053 |
| TScep. | 0.6252 ± 0.0106 | 0.4543 ± 0.0141 | 0.6665 ± 0.0032 | 0.3425 ± 0.0116 | 0.5199 ± 0.0046 | 0.2792 ± 0.0062 |
| STTran. | 0.5816 ± 0.0100 | 0.4334 ± 0.0133 | 0.7514 ± 0.0030 | 0.5046 ± 0.0107 | 0.5284 ± 0.0037 | 0.2972 ± 0.0050 |
| BIOT | 0.5139 ± 0.0099 | 0.3898 ± 0.0132 | 0.5613 ± 0.0029 | 0.1225 ± 0.0105 | 0.4595 ± 0.0046 | 0.1843 ± 0.0063 |
| EEGPT | 0.5279 ± 0.0123 | 0.4617 ± 0.0164 | 0.7323 ± 0.0026 | 0.4624 ± 0.0093 | 0.5231 ± 0.0044 | 0.2031 ± 0.0060 |
| LaBraM | 0.6234 ± 0.0125 | 0.4599 ± 0.0166 | 0.7874 ± 0.0032 | 0.5765 ± 0.0116 | 0.5414 ± 0.0037 | 0.3108 ± 0.0050 |
| CBraMod | 0.6139 ± 0.0117 | 0.4385 ± 0.0156 | 0.7895 ± 0.0028 | 0.5828 ± 0.0101 | 0.5454 ± 0.0042 | 0.3145 ± 0.0057 |
| ELASTIQ | 0.6381 ± 0.0120 | 0.4692 ± 0.0160 | 0.8134 ± 0.0031 | 0.6127 ± 0.0112 | 0.5483 ± 0.0038 | 0.3187 ± 0.0057 |

| Methods | SHU-MI | | HighGamma | | Cho2017 | |
| --- | --- | --- | --- | --- | --- | --- |
| | B.Acc | Kappa | B.Acc | Kappa | B.Acc | Kappa |
| EEGNet | 0.5665 ± 0.0007 | 0.1976 ± 0.0015 | 0.7856 ± 0.0038 | 0.5838 ± 0.0076 | 0.7644 ± 0.0062 | 0.5310 ± 0.0124 |
| Conformer | 0.5780 ± 0.0008 | 0.1591 ± 0.0018 | 0.7637 ± 0.0036 | 0.5627 ± 0.0071 | 0.7838 ± 0.0057 | 0.5719 ± 0.0115 |
| TScep. | 0.5564 ± 0.0007 | 0.2047 ± 0.0015 | 0.6851 ± 0.0031 | 0.5129 ± 0.0061 | 0.7339 ± 0.0056 | 0.4640 ± 0.0112 |
| STTran. | 0.5714 ± 0.0008 | 0.2191 ± 0.0018 | 0.7019 ± 0.0028 | 0.5162 ± 0.0056 | 0.7612 ± 0.0059 | 0.5260 ± 0.0117 |
| BIOT | 0.5587 ± 0.0008 | 0.2151 ± 0.0016 | 0.5824 ± 0.0035 | 0.4855 ± 0.0069 | 0.5413 ± 0.0055 | 0.0729 ± 0.0110 |
| EEGPT | 0.5228 ± 0.0009 | 0.1657 ± 0.0019 | 0.6516 ± 0.0038 | 0.5152 ± 0.0075 | 0.7197 ± 0.0058 | 0.4305 ± 0.0115 |
| LaBraM | 0.6338 ± 0.0008 | 0.2338 ± 0.0016 | 0.6939 ± 0.0036 | 0.5200 ± 0.0072 | 0.7614 ± 0.0053 | 0.5221 ± 0.0106 |
| CBraMod | 0.6403 ± 0.0007 | 0.2384 ± 0.0015 | 0.7684 ± 0.0034 | 0.5671 ± 0.0068 | 0.7439 ± 0.0059 | 0.4974 ± 0.0117 |
| ELASTIQ | 0.6134 ± 0.0008 | 0.2267 ± 0.0017 | 0.7982 ± 0.0035 | 0.5965 ± 0.0070 | 0.7908 ± 0.0060 | 0.5816 ± 0.0120 |

| Methods | Shin2017A | | PhysioNet-MI | | FACED | |
| --- | --- | --- | --- | --- | --- | --- |
| | B.Acc | Kappa | B.Acc | Kappa | B.Acc | Kappa |
| EEGNet | 0.7054 ± 0.0085 | 0.4521 ± 0.0171 | 0.6953 ± 0.0029 | 0.3980 ± 0.0177 | 0.4271 ± 0.0019 | 0.3512 ± 0.0357 |
| Conformer | 0.6464 ± 0.0071 | 0.2901 ± 0.0142 | 0.6951 ± 0.0034 | 0.3952 ± 0.0208 | 0.4943 ± 0.0019 | 0.4263 ± 0.0350 |
| TScep. | 0.5992 ± 0.0089 | 0.1891 ± 0.0179 | 0.6649 ± 0.0030 | 0.3216 ± 0.0182 | 0.2056 ± 0.0022 | 0.1088 ± 0.0410 |
| STTran. | 0.6168 ± 0.0093 | 0.2371 ± 0.0186 | 0.6706 ± 0.0035 | 0.3370 ± 0.0217 | 0.3791 ± 0.0023 | 0.2999 ± 0.0438 |
| BIOT | 0.5422 ± 0.0094 | 0.0909 ± 0.0188 | 0.4874 ± 0.0030 | 0.0162 ± 0.0184 | 0.1711 ± 0.0022 | 0.0647 ± 0.0407 |
| EEGPT | 0.5389 ± 0.0074 | 0.0688 ± 0.0148 | 0.6820 ± 0.0029 | 0.3730 ± 0.0180 | 0.3346 ± 0.0025 | 0.2486 ± 0.0469 |
| LaBraM | 0.6744 ± 0.0073 | 0.3607 ± 0.0145 | 0.7246 ± 0.0028 | 0.4487 ± 0.0174 | 0.5457 ± 0.0021 | 0.4809 ± 0.0393 |
| CBraMod | 0.6861 ± 0.0072 | 0.3689 ± 0.0144 | 0.7219 ± 0.0037 | 0.4386 ± 0.0229 | 0.5787 ± 0.0021 | 0.4941 ± 0.0385 |
| ELASTIQ | 0.7256 ± 0.0088 | 0.4811 ± 0.0176 | 0.6992 ± 0.0035 | 0.3983 ± 0.0214 | 0.5819 ± 0.0023 | 0.5243 ± 0.0430 |

| Methods | SEED | | SEED-IV | | SEED-V | |
| --- | --- | --- | --- | --- | --- | --- |
| | B.Acc | Kappa | B.Acc | Kappa | B.Acc | Kappa |
| EEGNet | 0.5337 ± 0.0122 | 0.3143 ± 0.0251 | 0.3651 ± 0.0102 | 0.1578 ± 0.0143 | 0.2932 ± 0.0102 | 0.1136 ± 0.0143 |
| Conformer | 0.6254 ± 0.0125 | 0.4342 ± 0.0186 | 0.4094 ± 0.0112 | 0.2121 ± 0.0174 | 0.3060 ± 0.0112 | 0.1335 ± 0.0174 |
| TScep. | 0.6369 ± 0.0155 | 0.4604 ± 0.0289 | 0.4063 ± 0.0078 | 0.1876 ± 0.0139 | 0.3637 ± 0.0078 | 0.1984 ± 0.0139 |
| STTran. | 0.5882 ± 0.0079 | 0.3873 ± 0.0133 | 0.3616 ± 0.0072 | 0.1415 ± 0.0121 | 0.2244 ± 0.0072 | 0.0344 ± 0.0121 |
| BIOT | 0.6674 ± 0.0118 | 0.5034 ± 0.0254 | 0.4141 ± 0.0187 | 0.1937 ± 0.0262 | 0.3045 ± 0.0187 | 0.1306 ± 0.0262 |
| EEGPT | 0.5054 ± 0.0105 | 0.2659 ± 0.0191 | 0.3202 ± 0.0144 | 0.0861 ± 0.0210 | 0.2253 ± 0.0144 | 0.0335 ± 0.0210 |
| LaBraM | 0.7083 ± 0.0107 | 0.5613 ± 0.0188 | 0.4415 ± 0.0138 | 0.2560 ± 0.0209 | 0.4010 ± 0.0138 | 0.2563 ± 0.0209 |
| CBraMod | 0.7102 ± 0.0089 | 0.5868 ± 0.0122 | 0.4605 ± 0.0097 | 0.2728 ± 0.0143 | 0.4029 ± 0.0138 | 0.2570 ± 0.0209 |
| ELASTIQ | 0.7011 ± 0.0068 | 0.5543 ± 0.0176 | 0.4630 ± 0.0034 | 0.2754 ± 0.0214 | 0.4126 ± 0.0023 | 0.2575 ± 0.0430 |

| Methods | SEED-VII | | OpenBMI-SSVEP | | eldBETA | |
| --- | --- | --- | --- | --- | --- | --- |
| | B.Acc | Kappa | B.Acc | Kappa | B.Acc | Kappa |
| EEGNet | 0.2587 ± 0.0010 | 0.1413 ± 0.0345 | 0.9430 ± 0.0036 | 0.9253 ± 0.0265 | 0.5922 ± 0.0036 | 0.5410 ± 0.0302 |
| Conformer | 0.3209 ± 0.0012 | 0.2081 ± 0.0439 | 0.8981 ± 0.0045 | 0.8634 ± 0.0337 | 0.5329 ± 0.0038 | 0.4736 ± 0.0320 |
| TScep. | 0.3300 ± 0.0011 | 0.2198 ± 0.0394 | 0.8026 ± 0.0043 | 0.7375 ± 0.0322 | 0.4205 ± 0.0030 | 0.3476 ± 0.0255 |
| STTran. | 0.1867 ± 0.0011 | 0.0516 ± 0.0400 | 0.9398 ± 0.0046 | 0.9195 ± 0.0341 | 0.4848 ± 0.0033 | 0.4199 ± 0.0283 |
| BIOT | 0.3096 ± 0.0011 | 0.1939 ± 0.0402 | 0.7936 ± 0.0040 | 0.7258 ± 0.0297 | 0.5476 ± 0.0033 | 0.4901 ± 0.0282 |
| EEGPT | 0.1809 ± 0.0013 | 0.0476 ± 0.0470 | 0.9301 ± 0.0047 | 0.9098 ± 0.0351 | 0.5821 ± 0.0034 | 0.5292 ± 0.0288 |
| LaBraM | 0.3244 ± 0.0011 | 0.2159 ± 0.0382 | 0.8698 ± 0.0041 | 0.8269 ± 0.0302 | 0.3653 ± 0.0032 | 0.2848 ± 0.0270 |
| CBraMod | 0.3311 ± 0.0010 | 0.2233 ± 0.0362 | 0.9205 ± 0.0039 | 0.8924 ± 0.0291 | 0.5817 ± 0.0038 | 0.5299 ± 0.0326 |
| ELASTIQ | 0.3356 ± 0.0012 | 0.2275 ± 0.0430 | 0.9462 ± 0.0043 | 0.9283 ± 0.0320 | 0.6262 ± 0.0035 | 0.5801 ± 0.0297 |

Table 7: Performance comparison on datasets (Balanced Accuracy and Cohen's Kappa) *(continued)*.

| Methods | Benchmark | | BETA | | BCIC-Speech | |
|---|---|---|---|---|---|---|
| | B.Acc | Kappa | B.Acc | Kappa | B.Acc | Kappa |
| EEGNet | $0.8018 \pm 0.0064$ | $0.7969 \pm 0.0422$ | $0.6291 \pm 0.0035$ | $0.6193 \pm 0.0337$ | $0.2699 \pm 0.0023$ | $0.0880 \pm 0.0413$ |
| Conformer | $0.7691 \pm 0.0062$ | $0.7635 \pm 0.0406$ | $0.4710 \pm 0.0030$ | $0.4574 \pm 0.0295$ | $0.4170 \pm 0.0019$ | $0.2722 \pm 0.0330$ |
| TScep. | $0.3329 \pm 0.0057$ | $0.3161 \pm 0.0374$ | $0.1911 \pm 0.0031$ | $0.1704 \pm 0.0307$ | $0.5314 \pm 0.0024$ | $0.4164 \pm 0.0426$ |
| STTran. | $0.7902 \pm 0.0056$ | $0.7850 \pm 0.0369$ | $0.5419 \pm 0.0028$ | $0.5303 \pm 0.0274$ | $0.4266 \pm 0.0019$ | $0.2832 \pm 0.0347$ |
| BIOT | $0.4771 \pm 0.0066$ | $0.4637 \pm 0.0431$ | $0.3358 \pm 0.0033$ | $0.3187 \pm 0.0317$ | $0.2920 \pm 0.0024$ | $0.1138 \pm 0.0427$ |
| EEGPT | $0.6200 \pm 0.0071$ | $0.6105 \pm 0.0470$ | $0.5453 \pm 0.0031$ | $0.5265 \pm 0.0307$ | $0.2364 \pm 0.0022$ | $0.0478 \pm 0.0397$ |
| LaBraM | $0.7692 \pm 0.0068$ | $0.7462 \pm 0.0446$ | $0.5572 \pm 0.0034$ | $0.5395 \pm 0.0328$ | $0.4819 \pm 0.0024$ | $0.3863 \pm 0.0434$ |
| CBraMod | $0.7436 \pm 0.0066$ | $0.7379 \pm 0.0431$ | $0.5239 \pm 0.0034$ | $0.5116 \pm 0.0328$ | $0.4280 \pm 0.0022$ | $0.2860 \pm 0.0395$ |
| ELASTIQ | $0.7702 \pm 0.0065$ | $0.7802 \pm 0.0427$ | $0.5483 \pm 0.0032$ | $0.5392 \pm 0.0312$ | $0.5453 \pm 0.0023$ | $0.4317 \pm 0.0410$ |

| Methods | ADHD-AliMotie | | Mental Workload | |
|---|---|---|---|---|
| | B.Acc | Kappa | B.Acc | Kappa |
| EEGNet | $0.6349 \pm 0.0256$ | $0.2789 \pm 0.0374$ | $0.5480 \pm 0.0224$ | $0.0982 \pm 0.0170$ |
| Conformer | $0.7316 \pm 0.0236$ | $0.4681 \pm 0.0344$ | $0.5984 \pm 0.0185$ | $0.1801 \pm 0.0140$ |
| TScep. | $0.7333 \pm 0.0252$ | $0.4752 \pm 0.0367$ | $0.6480 \pm 0.0177$ | $0.3034 \pm 0.0134$ |
| STTran. | $0.7515 \pm 0.0252$ | $0.5047 \pm 0.0368$ | $0.6335 \pm 0.0179$ | $0.2683 \pm 0.0136$ |
| BIOT | $0.6516 \pm 0.0223$ | $0.3155 \pm 0.0326$ | $0.5857 \pm 0.0222$ | $0.1480 \pm 0.0168$ |
| EEGPT | $0.7124 \pm 0.0208$ | $0.4217 \pm 0.0303$ | $0.4925 \pm 0.0192$ | $0.0223 \pm 0.0145$ |
| LaBraM | $0.6194 \pm 0.0200$ | $0.2285 \pm 0.0292$ | $0.5650 \pm 0.0181$ | $0.1643 \pm 0.0137$ |
| CBraMod | $0.6434 \pm 0.0253$ | $0.3121 \pm 0.0369$ | $0.5746 \pm 0.0200$ | $0.1695 \pm 0.0151$ |
| **ELASTIQ (Ours)** | $0.76990 \pm 0.0248$ | $0.52820 \pm 0.0362$ | $0.64930 \pm 0.0211$ | $0.31340 \pm 0.0160$ |

Table 8: Five-fold cross-validation performance comparison between ELASTIQ and CBraMod across three datasets. Results are reported as mean $\pm$ std. Statistical significance is assessed using the Wilcoxon signed-rank test ($^*$: $p < 0.05$, $^{**}$: $p < 0.01$).

| Method | OpenBMI-MI | | FACED | | OpenBMI-SSVEP | |
|---|---|---|---|---|---|---|
| | B.Acc | p-value | B.Acc | p-value | B.Acc | p-value |
| CBraMod | $0.7840 \pm 0.0272$ | – | $0.5510 \pm 0.0138$ | – | $0.9124 \pm 0.0253$ | – |
| ELASTIQ | $\mathbf{0.8270 \pm 0.016}^{**}$ | 0.0072 | $\mathbf{0.5680 \pm 0.029}^{*}$ | 0.0435 | $\mathbf{0.9250 \pm 0.013}^{*}$ | 0.0232 |

# D  ZERO-SHOT EVALUATION

To further assess whether instruction tuning enables ELASTIQ to generalize beyond standard cross-dataset settings, we additionally evaluate the model under a zero-shot regime. In this experiment, we hold out one entire dataset as unseen, train ELASTIQ on the remaining 19 datasets using multi-task instruction tuning, and directly evaluate the model on the excluded dataset without any dataset-specific adaptation. To isolate the contribution of instruction tuning itself, we compare ELASTIQ against a randomly initialized version of the same architecture under the identical zero-shot protocol. As shown in Table 9, ELASTIQ consistently and substantially outperforms the random-initialization baseline across all held-out datasets. For instance, on BCIC-IV2a (2-class), ELASTIQ improves the balanced accuracy to 0.5712 and Kappa to 0.1424. Similar improvements are observed on BCIC-IV2a (4-class) ($0.2500 \rightarrow 0.2804$), SEED ($0.3300 \rightarrow 0.3567$), SEED-IV ($0.2500 \rightarrow 0.2849$), SEED-V ($0.2000 \rightarrow 0.2503$), and SEED-VII ($0.1428 \rightarrow 0.1604$), with corresponding Kappa gains across all datasets. These results demonstrate that instruction-guided alignment transfers effectively to unseen datasets and does not rely on within-dataset fine-tuning. Notably, such a comprehensive zero-shot evaluation protocol has not been conducted in prior EEG foundation models, making ELASTIQ the first to systematically validate cross-dataset generalization under a strict no-adaptation scenario.

Table 9: Performance comparison between random initialization and zero-shot settings.

| Dataset | Random Init | | ELASTIQ | |
|---|---|---|---|---|
| | B.Acc | Kappa | B.Acc | Kappa |
| BCIC_IV2a (2-class) | 0.5000 | 0.0000 | 0.5712 | 0.1424 |
| BCIC_IV2a (4-class) | 0.2500 | 0.0000 | 0.2804 | 0.0405 |
| SEED | 0.3300 | 0.0000 | 0.3567 | 0.0462 |
| SEED-IV | 0.2500 | 0.0000 | 0.2849 | 0.0444 |
| SEED-V | 0.2000 | 0.0000 | 0.2503 | 0.0656 |
| SEED-VII | 0.1428 | 0.0000 | 0.1604 | 0.0193 |

## E   ANALYSIS OF PARAPHRASE ROBUSTNESS

To evaluate whether ELASTIQ depends on the exact linguistic phrasing of the task instruction, we conducted a paraphrase robustness study. For each task, we designed 10 semantically equivalent instructions that vary in grammatical structure, tone (imperative vs. declarative), and surface wording while preserving the intended task meaning. We replaced the original instruction with each paraphrase and measured the direct-inference decoding performance. As shown in Table 10, ELASTIQ exhibits remarkably stable performance across all paraphrases. For the *motor imagery* task, accuracies on MI-OpenBMI vary only within a narrow range of 0.7165–0.7250 for B.Acc and 0.4329–0.4500 for Kappa, with an overall variance below 0.25%. A similarly small fluctuation is observed on MI-Cho2017 (B.Acc 0.7306–0.7361; Kappa 0.4611–0.4722). For *emotion recognition*, the SEED dataset shows B.Acc tightly concentrated between 0.5602–0.5667 and Kappa between 0.3481–0.3588, while SEED-IV shows minimal spread around 0.3811–0.3838 for B.Acc and 0.2007–0.2048 for Kappa. Across both tasks, the standard deviations remain extremely small (e.g., MI-OpenBMI B.Acc STD = 0.0025, SEED B.Acc STD = 0.0045), confirming that instruction variations exert almost no influence on decoding performance. These findings demonstrate that ELASTIQ relies on the underlying semantic content of the instruction rather than the precise linguistic template, highlighting the robustness and flexibility of its instruction-conditioned alignment mechanism.

## F   ANALYSIS OF LEARNABLE QUERIES IN IQF

To better understand how the IQF influences our model performance, we vary the number of learnable queries $N_q \in \{4, 8, 12\}$ and evaluate model performance under the multi-task instruction tuning setting. As shown in Table 11, accuracy improves substantially from $N_q = 4$ to $N_q = 8$, but saturates when increasing $N_q$ to 12. This indicates that only a small set of queries is sufficient to capture instruction-conditioned EEG patterns and that adding more queries yields diminishing returns. These results support our design that the queries function as compact latent probes rather than requiring large capacity.

## G   EFFECT OF INCORRECT INSTRUCTIONS

To further examine the role of language guidance, we analyze cases where the model is deliberately given misleading instructions that do not match the underlying EEG dataset. Figure 5 shows examples on OpenBMI-MI and SEED-V datasets.

When provided with correct instructions, the learned feature spaces become more structured, with compact intra-class clusters and clearer inter-class separation. However, when misleading instructions are introduced, the feature space is distorted toward the semantics of the given instruction rather than the ground-truth task. For example, MI data conditioned on emotion-related instructions form clusters resembling affective categories, and SEED-V data prompted with MI or SSVEP instructions are reorganized into motor or frequency-based groupings. These results emphasize the strong controllability of our model through natural language. While correct instructions enhance discriminability, misleading instructions actively reshape the representation space according to the semantic

Table 10: Direct inference evaluation across instruction sentences for motor imagery and emotion tasks.

(a) Motor Imagery Task

| Instruction Sentence | MI-OpenBMI | | MI-Cho2017 | |
|---|---|---|---|---|
| | B.Acc | Kappa | B.Acc | Kappa |
| This is a motor imagery decoding task | 0.7190 | 0.4379 | 0.7356 | 0.4711 |
| Decode motor imagery | 0.7206 | 0.4413 | 0.7333 | 0.4667 |
| Please decode the motor imagery | 0.7208 | 0.4417 | 0.7361 | 0.4722 |
| Classify the motor imagery task | 0.7225 | 0.4450 | 0.7339 | 0.4678 |
| Identify the motor imagery class | 0.7223 | 0.4446 | 0.7356 | 0.4711 |
| Predict the type of imagined movement | 0.7206 | 0.4413 | 0.7328 | 0.4656 |
| Determine which motor imagery action is performed | 0.7244 | 0.4487 | 0.7344 | 0.4689 |
| Infer the imagined motor action from the EEG | 0.7250 | 0.4500 | 0.7306 | 0.4611 |
| Which motor imagery category does this trial belong to? | 0.7165 | 0.4329 | 0.7344 | 0.4689 |
| Analyze this EEG and classify the motor imagery | 0.7208 | 0.4417 | 0.7306 | 0.4611 |
| **Average ± STD** | 0.7213 ± 0.0025 | 0.4425 ± 0.0050 | 0.7337 ± 0.0019 | 0.4675 ± 0.0039 |

(b) Emotion Recognition Task

| Instruction Sentence | SEED | | SEED-IV | |
|---|---|---|---|---|
| | B.Acc | Kappa | B.Acc | Kappa |
| This is an emotion recognition task | 0.5602 | 0.3481 | 0.3779 | 0.1931 |
| Decode emotional states | 0.5663 | 0.3581 | 0.3835 | 0.2039 |
| Please decode the emotional states | 0.5643 | 0.3552 | 0.3838 | 0.2041 |
| Identify the emotional state | 0.5659 | 0.3574 | 0.3838 | 0.2048 |
| Classify the emotion represented in the signal | 0.5663 | 0.3579 | 0.3811 | 0.2007 |
| Predict the emotion category | 0.5656 | 0.3568 | 0.3831 | 0.2031 |
| Determine the emotional condition from this EEG | 0.5646 | 0.3553 | 0.3818 | 0.2008 |
| Infer the participant emotional state | 0.5667 | 0.3588 | 0.3836 | 0.2030 |
| What emotion does this EEG signal reflect? | 0.5655 | 0.3567 | 0.3818 | 0.2028 |
| Analyze the EEG and classify the emotion | 0.5662 | 0.3580 | 0.3818 | 0.2025 |
| **Average ± STD** | 0.5659±0.0045 | 0.3562±0.0070 | 0.3822±0.0018 | 0.2019±0.0034 |

| Number of Queries $N_q$ | MI-OpenBMI | | SEED | |
|---|---|---|---|---|
| | B.Acc (%) | Kappa | B.Acc (%) | Kappa |
| 4 | 0.6913 ± 0.0025 | 0.4125 ± 0.0050 | 0.7037 ± 0.0019 | 0.4375 ± 0.0039 |
| 8 | **0.7278 ± 0.0026** | **0.4481 ± 0.0048** | **0.7389 ± 0.0018** | 0.4675 ± 0.0039 |
| 12 | 0.7213 ± 0.0025 | 0.4425 ± 0.0050 | 0.7337 ± 0.0019 | **0.4692 ± 0.0037** |

Table 11: Direct Interference ablation study on the number of learnable queries $N_q$ evaluated on two datasets using Balanced Accuracy and Cohen's Kappa.

prior they provide. This highlights both the power and sensitivity of instruction-conditioned alignment in EEG-FMs.

## H  EFFECT OF LANGUAGE EMBEDDING MODEL

Language model plays an important role in ELASTIQ, as it defines the semantic space into which EEG representations are aligned; therefore, the choice of text encoder can substantially influence instruction effectiveness and downstream performance. To examine this effect, we evaluated several modern embedding models, namely sbert (all-mpnet-base-v2) (Reimers & Gurevych, 2019), bert-base-uncased (Devlin et al., 2019), E5-large-v2 (Wang et al., 2023), Qwen3-embedding-0.6B (Bai et al., 2024), Voyage-3-large (VoyageAI, 2024) as frozen text encoders in ELASTIQ. The results are obtained from training the model during the multi-task instruction tuning stage. As shown in Figure 6, all-mpnet-base-v2 achieves the best convergence and highest decoding accuracy. This is well aligned with its design: MPNet-based sbert models are explicitly optimized for short sentence-level semantic similarity, which matches the nature of ELASTIQ's task instructions and label phrases (e.g., "left hand", "happy", "12 Hz"). In contrast, E5-large-v2, Qwen3-embedding, and Voyage-3-large focus primarily on large-scale retrieval or multilingual embedding, which results in embedding geometries less suited for fine-grained semantic alignment of short instructions. BERT-base-

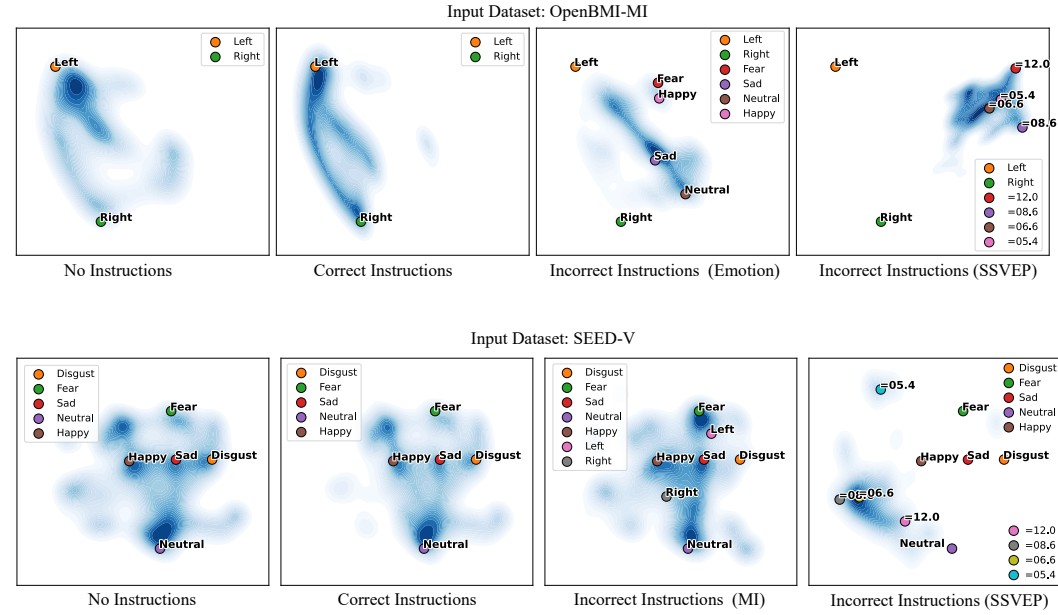

Figure 5: Comparison of KDE visualization of features between incorrect and correct instructions.

uncased, lacking sentence-level training, performs the weakest as expected. These results confirm that SBERT's compact and semantically stable embedding space is the most appropriate for ELAS-TIQ, supporting our design choice.

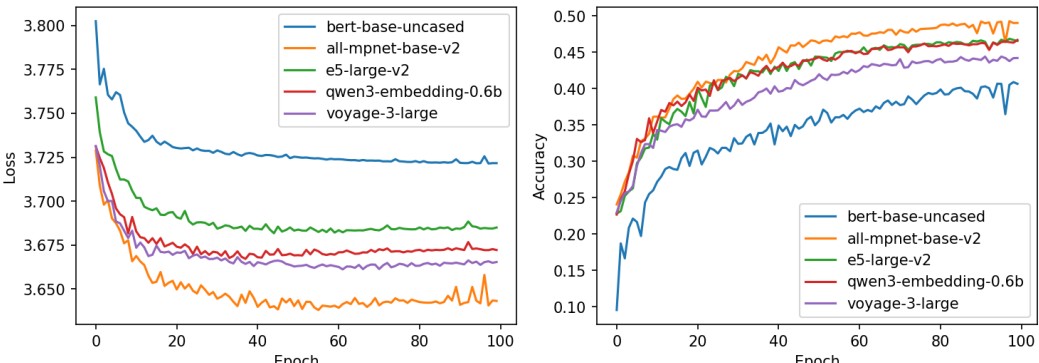

Figure 6: Comparison among different language models

# I EFFECT OF IQF AND TEXTUAL PROTOTYPES ON ALIGNMENT

To disentangle the respective contributions of the Instruction-conditioned Q-Former (IQF) and the semantic target prototypes, we performed additional ablation studies that isolate each component. Specifically, we evaluate two variants of ELASTIQ: (1) IQF without semantic targets, where instruction-conditioned modulation and query-based interaction are retained but textual label prototypes are removed and classification is performed using a learned linear head; (2) Semantic targets without IQF, where class-level text embeddings are preserved but the Q-Former is removed, and EEG features are directly aligned with label prototypes without query-based filtering.

As summarized in Table 12, removing either component leads to a clear degradation in balanced accuracy across datasets. This demonstrates that IQF and semantic target alignment each provide in-

dependent benefits, and their combination yields complementary improvements by jointly enabling semantic conditioning (via IQF) and cross-modal alignment (via text prototypes). This confirms that both IQF and semantic target contribute to he performance of ELASTIQ.

Table 12: Ablation on the contributions of IQF and semantic targets.

| IQF | Semantic Targets | OpenBMI-MI | | SEED | |
|-----|------------------|------------|-------|------|-------|
| | | B.Acc | Kappa | B.Acc | Kappa |
| − | ✓ | 0.7590 | 0.5302 | 0.6520 | 0.4923 |
| ✓ | − | 0.7912 | 0.5987 | 0.6328 | 0.4581 |
| ✓ | ✓ | **0.8144** | **0.6287** | **0.7011** | **0.5543** |

## J  TOKEN CLUSTERING VISUALIZATION FOR DUAL MASKING BRANCHES

We further visualize the feature learnt by the bidirectional transformer and the causal transformer. Results in Figure 7 yield clearly separable clusters, indicating that they extract distinct and complementary features from EEG signals. This validates the design of joint STR: by optimizing structural and temporal objectives separately, the model learns representations that capture different aspects of EEG dynamics, which together provide a richer and more transferable embedding.

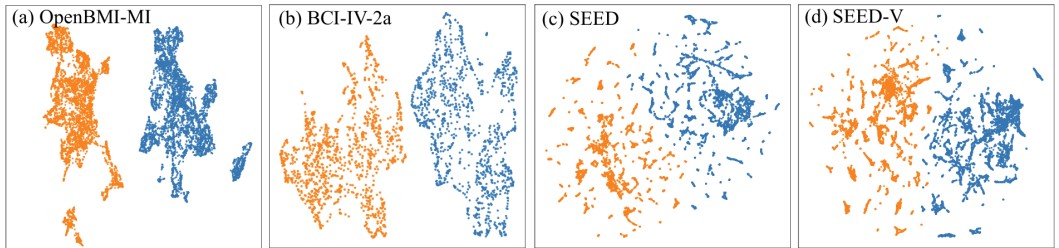

Figure 7: Token clustering after UMAP visualization. **Yellow points**: tokens from bidirectional transformer. **Blue points**: tokens from causal transformer

## K  PARAMETER SETTINGS

We list the hyperparameters selected in our model in Table 13, and training parameters in Table 14. Parameter size of ELASTIQ can be found at Table 15

## L  TARGET TEXTS FOR DOWNSTREAM DATASETS

We list the target texts for each downstream dataset in Table 16.

## M  PRE-TRAINING DATASET DESCRIPTION

- Stieger2021 (Stieger et al., 2021): This database contains EEG recordings from 62 healthy participants, each completing 7–11 sessions of BCI training to control a computer cursor in one- and two-dimensional spaces using motor imagery. Data were collected with 62 electrodes, and accompanying behavioral measures.
- SEED-FRA (Liu et al., 2022b): Eight French subjects participated in the experiments. Twenty-one film clips (positive, neutral and negative emotions) were chosen as stimuli used in the experiments.
- SEED-GER (Liu et al., 2022b): Eight German subjects participated in the experiments. Twenty film clips (positive, neutral and negative emotions) were chosen as stimuli used in the experiments.

Table 13: Hyperparameters for ELASTIQ

| Hyperparameters | Value |
| --- | --- |
| *Tokenization* | |
| Sampling rate | 200 hz |
| Segment window length | 0.5s |
| Input channels | 65 |
| Temporal kernel size | (1, 40), padding (1, 20) |
| Spatial kernel size | (65, 1) |
| Pooling | (1, 10) |
| *Transformers* | |
| Transformer encoder layers | 12 |
| Token size | 256 |
| Fead-forward scale | 4 |
| Attention head number | 8 |
| Dropout | 0.1 |
| Mask ratio | 0.5 |
| Frequency cutoff range | 1–50 Hz |
| Frequency cutoff Band | 6 Hz continuous band |
| *Q-Former* | |
| Number of queries | 8 |
| Q-Former layers | 4 |
| Text embedding dimension | 768 |
| Text embedding model | bert-base-uncased / all-mpnet-base-v2 |

Table 14: Training Parameters for ELASTIQ

| Hyperparameters | Value |
| --- | --- |
| Batch size | 512 |
| Peak learning rate | $1 \times 10^{-3}$ |
| LR scale (transformer params) | 0.1 |
| LR scale (other params) | 1.0 |
| Minimal learning rate | $1 \times 10^{-4}$ |
| Learning rate scheduler | Cosine Annealing lr |
| Optimizer | AdamW ($\beta = 0.9, 0.999$) |
| Weight decay | $1 \times 10^{-3}$ |
| Precision | bf16-mixed |

- SEED-SD (Li et al., 2025): SEED-SD is a multimodal EEG and eye-tracking dataset collected from 40 healthy participants under three sleep-related conditions—sleep deprivation, sleep recovery, and normal sleep. In each condition, participants watched 24 video clips (six per emotion) designed to evoke four basic emotions: happiness, sadness, fear, and neutral; each clip lasts about 2.5 minutes.

- ChineseEEG (Mou et al., 2024): This dataset consists EEG data recorded from 10 participants silently reading approximately 13 hours of Chinese text from two well-known novels.

- Chisco dataset (Zhang et al., 2024): The Chisco dataset is a large-scale EEG corpus collected from three subjects for imagined speech decoding, featuring over 20,000 sentences and more than 900 minutes of high-density EEG per subject. It covers 6,000+ everyday phrases across 39 semantic categories, with trials designed to include both reading and imagined speech phases.

- LargeSpanish (Valle et al., 2024): This dataset consists of 60 EEG sessions from 56 healthy participants. It is recorded using a 64-channel EEG system during speech perception and silent speech production tasks involving 30 daily-use sentences in Spanish.

- ThinkOutLoud (Nieto et al., 2022): This open-access EEG dataset comprises recordings from 10 participants, collected using a 136-channel system across three paradigms—inner speech, pronounced speech, and visualized condition.

Table 15: Parameter Size of ELASTIQ

| Hyperparameters | Value |
|---|---|
| Tokenizer | 0.27 M |
| Dual Transformer | 20.23 M |
| Q-Former | 6.19 M |
| Total trainable parameters | 26.42 M |

Table 16: Datasets and corresponding decoding targets.

| Dataset | Targets |
|---|---|
| OpenBMI-MI | Right, Left |
| BCIC-IV2a | Left, Right, Foot, Tongue |
| BCIC-Upperlimb | Cylin, Sphe, Lumbrical |
| SHU-MI | Right, Left |
| HighGamma | Left, Right, Foot |
| Cho2017 | Left, Right |
| Shin2017A | Left, Right |
| PhysioNet-MI | Left, Right |
| FACED | Anger, Fear, Disgust, Sad, Amusement, Inspiration, Joy, Tenderness, Neutral |
| SEED | Positive, Neutral, Negative |
| SEED-IV | Neutral, Sad, Fear, Happy |
| SEED-V | Disgust, Fear, Sad, Neutral, Happy |
| SEED-VII | Happy, Surprise, Neutral, Sad, Disgust, Fear, Anger |
| OpenBMI-SSVEP | 12.0, 08.6, 06.6, 05.4 |
| eldBETA | 08.0, 09.5, 11.0, 08.5, 10.0, 11.5, 09.0, 10.5, 12.0 |
| Benchmark | 40 freq. classes (8.0–15.8Hz, step 0.2Hz) |
| BETA | 40 freq. classes (8.0–15.8Hz, step 0.2Hz) |
| BCIC-Speech | hello, help-me, stop, thank-you, yes |
| ADHD-AliMotie | Healthy, ADHD |
| Workload | Resting, Workload |

## N  DETAILS ABOUT BASELINE IMPLEMENTATION

To ensure a fair comparison, all baseline models were re-trained or fine-tuned using their officially released implementations and recommended hyperparameters. For each dataset, EEG trials were resampled to 200 Hz, truncated to a fixed length of 200 samples (65 channels), and, for transformer-based foundation models, further segmented into non-overlapping 200-sample windows as tokens. We trained all models for 100 epochs using Adam with a learning rate of 1e-3 and cross-entropy loss, and we evaluated balanced accuracy, ROC-AUC, weighted F1, and Cohen's Kappa on the validation and test splits. Pretrained checkpoints of LaBraM, EEGPT, and CBraMod were loaded and then fine-tuned end-to-end from the official checkpoints. For LaBraM and NeuroLM, we adopted their base model. The direct comparison between NeuroLM in task-specific inference is unfair. Therefore, we use a direct inference setting by only use the ELASTIQ after multi-task instruction tuning to predict the test data.

## O  MORE DETAILS ABOUT EXPERIMENTAL SETTING ON DOWNSTREAM DATASETS

- BCIC-IV-2a dataset (Tangermann et al., 2012) comprises recordings from nine subjects, each participating in two sessions of a four-class MI paradigm (left hand, right hand, foot,

and tongue). EEG was collected using 22 scalp electrodes and three EOG channels at 250 Hz.

- OpenBMI-MI dataset (Lee et al., 2019) provides a large-scale benchmark for brain–computer interface research. Its MI subset contains data from 54 subjects, each participating in two sessions. Subjects performed left- and right-hand motor imagery tasks, with approximately 100 trials per session, recorded using a 64-channel EEG system at 1000 Hz.

- BCIC-Upperlimb dataset (Jeong et al., 2022) is from BCI Competition 2021 – Track 4. It provides EEG recordings of subjects performing three unilateral grasp movements (cylindrical, spherical, lumbrical) across three consecutive days (train/validation/test), designed to evaluate upper-limb movement decoding and session-to-session transfer.

- SHU-MI dataset (Yang et al., 2025) includes high-quality multi-day recordings from 62 participants. Fifty-one subjects performed a two-class MI paradigm (left vs. right hand grasping), while eleven subjects performed a three-class paradigm (left hand, right hand, and foot). Each participant contributed three sessions, with both raw and preprocessed EEG data publicly available.

- High-Gamma dataset (Schirrmeister et al., 2017) was collected at TU Berlin and contains 128-channel EEG recordings from 14 subjects. Participants performed four tasks (left hand, right hand, both feet, and rest). Each subject completed 13 runs, yielding approximately 1000 four-second trials. We select left and right MI as evaluation tasks.

- Cho2017 dataset (Cho et al., 2017) contains EEG recordings from 52 subjects performing four-class motor imagery tasks (left hand, right hand, foot, tongue) using a 62-channel montage at 1,000 Hz sampling rate.

- PhysioNet-MI (Schalk et al., 2004) is a publicly available dataset on PhysioNet. It comprises EEG recordings from 109 healthy subjects performing both motor execution and motor imagery tasks involving the left and right hands.

- Shin2017A (Shin et al., 2016) dataset contains EEG recordings from 30 healthy subjects (29 right-handed, 1 left-handed; average age $28.5 \pm 3.7$ years). Subjects performed two-class hand motor imagery tasks (left vs. right hand) using a 30-channel EEG montage at 1000 Hz. Each participant completed three sessions with 20 trials per session (10 per class).

- SEED dataset (Duan et al., 2013) contains EEG and eye movement data of 12 subjects and EEG data of another 3 subjects. Data was collected when they were watching film clips.

- SEED-IV (Zheng et al., 2018) contains data from 15 subjects, each undergoing three sessions. During each session, 24 movie clips were used to elicit four discrete emotions: happy, sad, fear, and neutral. EEG was recorded using a 62-channel NeuroScan system at 1000 Hz, along with synchronized eye-tracking signals.

- SEED-V (Liu et al., 2021) expands the categories to five (happy, sad, fear, disgust, and neutral) and includes recordings from 20 subjects, each with three sessions and 15 clips per session.

- SEED-VII (Jiang et al., 2024a) further extends to seven categories (happy, sad, fear, disgust, neutral, anger, and surprise), employing 80 video stimuli, and was recorded with EEG and Tobii Pro Fusion eye-tracking from 20 subjects.

- FACED dataset (Chen et al., 2023) includes EEG recordings from 123 healthy participants exposed to 28 film clips designed to induce nine fine-grained emotions: amusement, inspiration, joy, tenderness, anger, fear, disgust, sadness, and neutral. EEG was collected using a 32-channel cap (10–20 system) at 250 Hz. In addition to categorical labels, dimensional ratings such as valence, arousal, familiarity, and liking were provided.

- Benchmark (Wang et al., 2016) is one of the most widely adopted SSVEP corpora, consisting of 35 subjects with 64-channel EEG recordings. Participants performed a cued spelling task involving 40 visual targets driven by joint frequency and phase modulation within the 8–15.8 Hz range. This dataset has become a de facto standard for assessing algorithmic performance in high target-count speller systems.

- BETA (Liu et al., 2020) extends the Benchmark dataset by including 70 subjects under a similar 40-target spelling paradigm with 64-channel recordings. The larger subject pool

provides a solid basis for evaluating cross-subject transferability and generalization of SSVEP decoding algorithms.

- eldBETA (Liu et al., 2022a) focuses on the aging population, containing EEG data from 100 elderly participants (aged 52–81). Each subject completed a 9-target SSVEP task with 64 channels. This dataset enables the investigation of age-related changes in neural responses and the development of BCI systems tailored for elderly users.

- OpenBMI–SSVEP (Lee et al., 2019) is part of the OpenBMI dataset and provides recordings from 30 healthy adults across two sessions. The paradigm consisted of 4 visual targets presented at the screen edges, with stimulation frequencies of 5.45, 6.67, 8.57, and 12 Hz. The dataset is well-suited for studying low-frequency responses, small-class classification, and cross-session robustness.

- The BCIC2020-3 dataset (Jeong et al., 2022), released as part of the International BCI Competition 2020, contains multi-class imagined speech EEG recordings from 15 healthy subjects. Participants were instructed to imagine speaking five short phrases while 64-channel EEG signals were recorded, providing a benchmark resource for covert speech decoding research.

- ADHD-AliMotie (Nasrabadi et al., 2020) recruited 121 children, including 61 diagnosed with ADHD and 60 healthy controls. Participants were between 7 and 12 years old, comprising both boys and girls. EEG was recorded using 19 electrodes at a sampling rate of 128 Hz.

- Mental Workload (Zyma et al., 2019) contains 36 subjects performing serial subtraction. EEG was recorded using 19 electrodes at a sampling rate of 500 Hz.

  Please find the summary of downstream datasets in Table 17.

Table 17: Summary of downstream EEG datasets used in this study.

| Dataset | Task | #Classes | #Subjects | #Channels | Sampling |
|---|---|---|---|---|---|
| BCIC-IV-2a | MI | 4 | 9 | 22 | 250 Hz |
| OpenBMI-MI | MI | 2 | 54 | 64 | 1000 Hz |
| BCIC-Upperlimb | MI | 3 | 9 | 22 | 250 Hz |
| SHU-MI | MI | 2 | 62 | 64 | 250 Hz |
| High Gamma | MI | 3 | 14 | 128 | 1000 Hz |
| Cho2017 | MI | 2 | 62 | 64 | 1000 Hz |
| Shin2017A | MI | 2 | 62 | 64 | 1000 Hz |
| PhysioNet | MI | 2 | 62 | 64 | 160 Hz |
| SEED | Emotion | 4 | 15 | 62 | 1000 Hz |
| SEED-IV | Emotion | 4 | 15 | 62 | 1000 Hz |
| SEED-V | Emotion | 5 | 20 | 62 | 1000 Hz |
| SEED-VII | Emotion | 7 | 20 | 62 | 1000 Hz |
| FACED | Emotion | 9 | 123 | 32 | 250 Hz |
| OpenBMI–SSVEP | SSVEP | 4 | 30 | 64 | 1000 Hz |
| BETA | SSVEP | 40 | 70 | 64 | 1000 Hz |
| eldBETA | SSVEP | 9 | 100 | 64 | 1000 Hz |
| Benchmark | SSVEP | 40 | 35 | 64 | 1000 Hz |
| BCIC2020-3 | Covert speech | 5 | 22 | 23 | 256 Hz |
| ADHD-AliMotie | Healthcare | 2 | 121 | 64 | 128 Hz |
| Mental Workload | Healthcare | 2 | 36 | 64 | 500 Hz |

## P  METRICS FOR QUANTIFYING REPRESENTATION SHIFT AND SEMANTIC ALIGNMENT

To quantitatively assess how natural-language instructions and label semantics reshape the EEG representation space, we employ several metrics that capture intra-class compactness, inter-class sep-

aration, and alignment with textual label prototypes. This section provides their formal definitions and interpretive meanings.

**Intra-class variance.** The intra-class variance characterizes the compactness of samples belonging to the same category. For class $c$ with embedding vectors $\{z_i\}_{i \in C_c}$ and class mean $\mu_c$, it is defined as

$$\text{Intra} = \frac{1}{K} \sum_{c=1}^{K} \frac{1}{|C_c|} \sum_{i \in C_c} \|z_i - \mu_c\|_2^2, \tag{14}$$

where $K$ is the number of classes. Smaller values indicate tighter class clusters and higher within-class consistency, while larger values reflect more dispersed and less coherent representations.

**Inter-class distance.** To quantify global separation among categories, we compute the average pairwise distance between class means:

$$\text{Inter} = \frac{2}{K(K-1)} \sum_{c<d} \|\mu_c - \mu_d\|_2. \tag{15}$$

Larger values imply well-separated categories with stronger discriminative structure in the embedding space, whereas smaller values indicate overlapping or weakly separated classes.

**Semantic margin.** We measure semantic discriminability by directly contrasting the cosine similarity to the correct prototype and the highest similarity to any incorrect prototype:

$$\text{Margin}_i = \frac{\langle z_i, e_{y_i} \rangle}{\|z_i\|_2 \|e_{y_i}\|_2} - \max_{c \neq y_i} \frac{\langle z_i, e_c \rangle}{\|z_i\|_2 \|e_c\|_2}. \tag{16}$$

Larger margins indicate stronger semantic preference for the correct class, whereas small or negative margins reflect confusion with competing prototypes.

## Q    EVALUATION SETTINGS ON DOWNSTREAM DATASETS

We report the details about the evaluation settings for all downstream datasets in Table 18.

## R    VISUALIZATION OF PRETRAINING LOSS

Figure 8 plots the loss trajectories for the bidirectional (random masking) and causal (future masking) transformers during pretraining. Both objectives decrease monotonically, confirming effective optimization. However, the causal transformer converges more rapidly, reaching a stable minimum around epoch 25 with a lower final loss. In contrast, the bidirectional transformer converges more slowly and plateaus at a higher loss. This divergence may be explained by the fact that causal prediction imposes stronger sequential constraints that accelerate convergence, whereas bidirectional reconstruction likely requires integrating information across the entire context, making optimization more challenging.

## S    STANDARD EEG MONTAGE USED BY ELASTIQ

In our framework, heterogeneous EEG datasets with different channel montages were spatially aligned onto a standard 10–10 electrode layout, as illustrated in Figure 9. Each electrode in the datasets was mapped to the nearest neighbor on this template, thereby preserving the spatial topology of the scalp distribution. This alignment ensures consistent channel representation across datasets and facilitates effective modeling of spatial dependencies.

## T    TOPOGRAPHY VISUALIZATION

Figure 10 presents saliency maps derived from our model across three representative tasks, highlighting the EEG components most influential for prediction. For motor imagery (OpenBMI-MI),

Table 18: Train/validation/test split strategies for downstream EEG datasets.

| Dataset | Split Strategy |
|---|---|
| BCIC-IV-2a | Multi-subject: For each subject, first 75% trials train/val, last 25% test; 20% of train/val for validation. |
| OpenBMI-MI | Cross-subject: Subjects 1–42 for train/val, 43–54 for test; 20% of train/val for validation. |
| BCIC-Upperlimb | Multi-subject (Same as BCIC-IV-2a) |
| SHU-MI | Multi-subject (Same as BCIC-IV-2a) |
| HighGamma | Multi-subject (Same as BCIC-IV-2a) |
| Cho2017 | Cross-subject: Subjects (total 49) 1–40 for train/val, 41–49 for test; 20% of train/val for validation. |
| Shin2017A | Multi-subject (Same as BCIC-IV-2a) |
| PhysioNet-MI | Cross-subject: Subjects 1–80 for train/val, 81–109 for test; 20% of train/val for validation. |
| SEED | Multi-subject Trial-based: For each subject (15), trials 1–9 train, 10–12 val, 13–15 test. |
| SEED-IV | Multi-subject Trial-based: For each subject (15), trials 1–16 train, 17–20 val, 21–24 test. |
| SEED-V | Multi-subject Trial-based: For each subject (16), trials 1–5 train, 6–10 val, 11–15 test. |
| SEED-VII | Multi-subject Trial-based: For each subject (20), trials 1–10 train, 11–15 val, 16–20 test. |
| FACED | Cross-subject: Subjects 1–100 train/val, 101–122 test; 20% of train/val for validation. |
| OpenBMI-SSVEP | Cross-subject: Subjects 1–42 train/val, 43–54 test; 20% of train/val for validation. |
| BETA | Cross-subject: Subjects 1–46 train, 47–50 val, 51–70 test. |
| eldBETA | Cross-subject: Subjects 1–75 train, 76–80 val, 81–100 test. |
| Benchmark | Cross-subject: Subjects 1–26 train, 27–28 val, 29–35 test. |
| BCIC-Speech | Multi-subject: First 300 trials train, remaining trials test (validation drawn from training set). |
| ADHD-AliMotie | Cross-subject: 70 subjects (35 ADHD + 35 controls) train, 10 (5+5) val, 40 (20+20) test. |
| Mental Workload | Cross-subject: Subjects 0–31 train/val, 32–35 test; 20% of train/val for validation. |

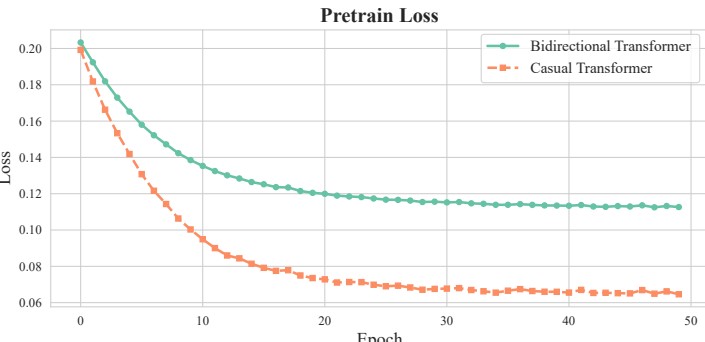

Figure 8: Pretraining loss of the bidirectional transformer with random masking and the causal transformer with next token masking

the model highlights contralateral motor cortex regions around C3 and C4 when distinguishing left-versus right-hand movements, consistent with established neurophysiological findings. For emotion recognition (SEED), salient activations emerge in frontal and temporal regions, reflecting neural substrates involved in affective processing. For SSVEP (Benchmark), the maps exhibit strong responses over occipital areas, in line with the visual cortex origin of steady-state responses. These results confirm that our model captures task-relevant neural patterns, thereby improving interpretability and supporting the neuroscientific plausibility of the learned representations.

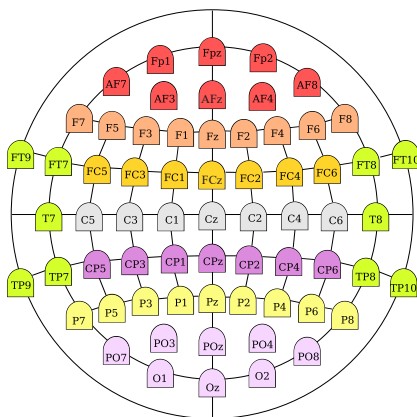

Figure 9: ELASTIQ employs the 10–10 system with 65 EEG electrodes; any input montage is interpolated to this configuration before being fed into ELASTIQ.

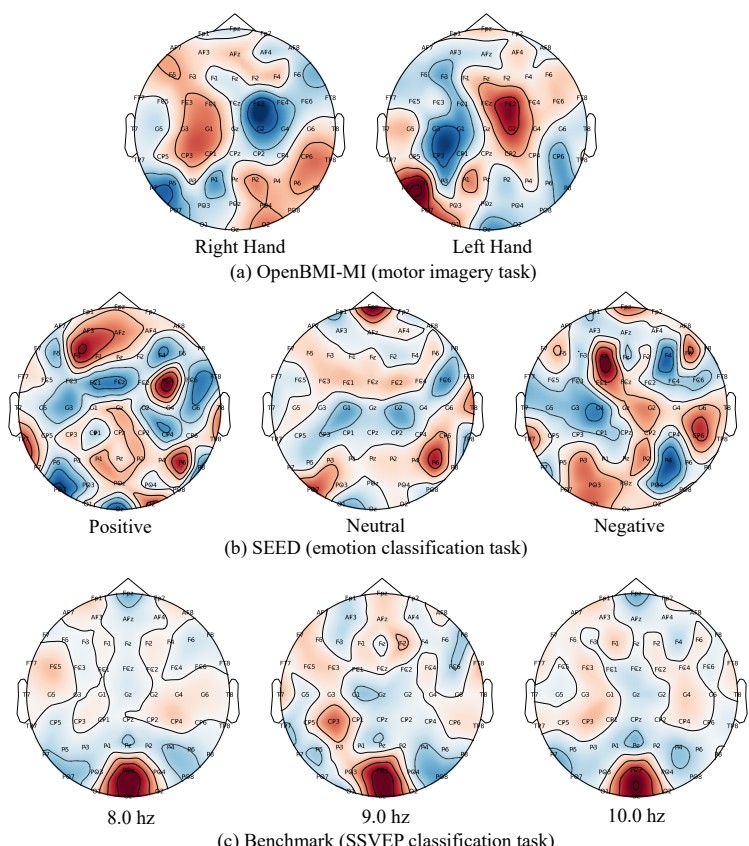

(a) OpenBMI-MI (motor imagery task)

(b) SEED (emotion classification task)

(c) Benchmark (SSVEP classification task)

Figure 10: Topography visualization on downstream datasets

## U    REPRODUCIBILITY STATEMENT

We have made every effort to ensure the reproducibility of our work. The training and evaluation pipelines are described in detail in Appendix Q, including dataset preprocessing, model configurations, and evaluation protocols. All datasets used are publicly available, and the code, pretrained checkpoints, and scripts for data preprocessing and evaluation will be fully released upon publication.

## V  THE USE OF LARGE LANGUAGE MODELS (LLMs)

Large language models (LLMs) were exclusively used to refine the writing of this manuscript, such as improving grammar, clarity, and readability. They were not involved in generating scientific content, designing experiments, or interpreting results. All research ideas, technical contributions, and analyses presented in this paper were conceived, implemented, and validated entirely by the authors.

