# OpenReview forum: "ELASTIQ: EEG–Language Alignment with Semantic Task Instruction and Querying"
_ICLR.cc/2026/Conference — ICLR 2026 Conference Desk Rejected Submission_

### Official Review · Reviewer_KRrL · 2025-10-15

**Soundness:** 2
**Presentation:** 3
**Contribution:** 2
**Rating:** 4
**Confidence:** 4

**Summary:**

This work introduces ELASTIQ: a new foundation model connecting brain activity and language representations.
It is composed of two stages:
- Spectro-Temporal Reconstruction (STR): a self-supervised learning task where the model reconstructs frequency-masked EEG signals (bidirectional and next token) to capture frequency and temporal structure.
- Instruction-based Query Fusion (IQF): at fine-tuning time, textual instructions are injected into EEG representations via FiLM conditioning and a Q-Former. This allows the model to align EEG embeddings with text embeddings in order to steer the brain features through language.

ELASTIQ is tested on 20 datasets across 5 families of tasks and achieves SOTA results on 14/20 datasets.

**Strengths:**

- The combination of STR and IQF is well explained and technically sound.
- The evaluation is broad, across 20 datasets, and the consistent performance improvement across different paradigms supports generalization.
- Language alignment seems to affect predictions deeply. This is not a superficial multimodal trick.
- ELASTIQ outperforms larger models like NeuroLM despite being compact.
- Multiple ablations and analyses are performed, giving credibility to the different design choices.

**Weaknesses:**

- Statistical Rigor

The authors do not mention how well they tune their baseline models. There is also a lack of statistical tests.
All tables including the main one (Table 1) do not show any information about variance/standard deviation across multiple seeds.
This is problematical especially for little gains like the one given by the Informed Task & Targets (Table 2).
The only table including infos on standard deviation is Table 5 (comparison with NeuroLM). This point is a critical weakness of the work.

- Language Prior Fairness

Another concern is about the model's use of pretrained text embeddings. It injects strong semantic knowledge (e.g. "left hand") which baseline models do not have.
Without baselines using similar priors, it's hard to understand how much of the improvement comes from EEG modeling.

- Instruction Robustness

Instruction robustness is not deeply tested: small variations in the instructions could change results.
We don't know how sensitive the model is to the given instruction.

I’d be happy to revisit my rating should the authors address the points raised in this review.

**Questions:**

- How can you prove that the model is not "borrowing" from the language model? Did you implement text-prototype classification for baselines to control for language priors? How to isolate gains due to IQF and gains due to semantic targets?

- How sensitive are results to paraphrases or synonyms? Can you report variance over 10 randomized paraphrases per task?

- What's the performance if all 20 datasets are evaluated with unseen subjects ? Why changing the splitting rule using an arbitrary rule like 30 subjects?

- Can you systematically report statistical test results and use cross-validation for each result?

- Any interesting results on the queries? Importance of their numbers? Any Specialization?

---

> ### Author Response · Authors · 2025-11-25
>
> We thank the reviewer for the positive review and insightful analysis.
>
> ---
> **[W1-1]**: The authors do not mention how well they tune their baseline models.
>
> **[R]**: Thank you so much for pointing out the missing clarification regarding how baseline models are tuned. In response, we have added a dedicated section in the appendix that systematically explains the training and fine-tuning procedures used for all baselines. In this new section, we describe the preprocessing pipeline applied consistently across datasets, the training configurations used for both conventional models and foundation models, and how pretrained checkpoints were handled to ensure consistency with their official implementations. We also explain how we align the evaluation protocol across baselines, including metrics and dataset splits. In addition, we clarify that to avoid an unfair comparison with NeuroLM, which does not perform task-specific fine-tuning, we also evaluate ELASTIQ under a direct inference setting where the instruction-tuned model is applied to downstream datasets without additional adaptation.
>
> Please find the details about the added experiment at Line 1170-1182, Page 22.
>
> We hope this explanation and additional experiment will provide a clear and fair evaluation of our method in comparison with existing baselines. Thank you again for your thoughtful feedback.
>
> ---
> **[W1-2+Q3]**: There is also a lack of statistical tests. All tables including the main one (Table 1) do not show any information about variance/standard deviation across multiple seeds. This is problematical especially for little gains like the one given by the Informed Task & Targets (Table 2). The only table including infos on standard deviation is Table 5 (comparison with NeuroLM). This point is a critical weakness of the work.
> Can you systematically report statistical test results and use cross-validation for each result?
>
> **[R]**: Thank you so much for this valuable feedback. In response, we have rerun all experiments reported in Table 1 using three random seeds, and we now report the mean across seeds. For Table 1, we keep the original random-seed evaluation protocol to remain consistent with prior work, but due to space limitations we report only the mean value in the main text. Kindly please refer to Line 375-421, Page 7 for the tables. The complete results, including both mean and standard deviation, are now provided in the Appendix B (Line 730-736, Page 14). For table 2, the result comes from direct inference, where the ELASTIQ is direct test on downstream dataset without finetune. So, the results produce no variance or standard deviation. To further assess the statistical reliability of the improvements, we conducted cross-validation experiments to compare our method with the best-performing baseline and performed pairwise Wilcoxon signed-rank tests to confirm the significance.
>
> Kindly please refer to line 740-755, Page 14 for the detail of the results.
>
> We hope our explanation will adress your concern. Thank again for your valuable feedback.
>
> ---
> **[W2]**: Another concern is about the model's use of pretrained text embeddings. It injects strong semantic knowledge (e.g. "left hand") which baseline models do not have. Without baselines using similar priors, it's hard to understand how much of the improvement comes from EEG modeling.
>
> **[R]**: Thank you so much for this valuable comment. Following your suggestion, we selected NeuroLM as the EEG-language baseline. We conducted a direct inference experiment in which ELASTIQ is evaluated without any task-specific fine-tuning. In the revised manuscript, we clarify that the model is kept frozen after the multi-task instruction tuning stage and is directly applied to each downstream dataset. This enables a unified evaluation protocol similar to that used by NeuroLM. We have also added a discussion comparing the two models, emphasizing the differences in their alignment mechanisms and prediction strategies, and explaining how these distinctions influence performance under direct inference.
>
> Kindly refer to Line 513-528, Page 10.
>
> We hope this explanation and additional experiment provide a relatively fair comparison between ELASTIQ and baseline using semantic knowledge. Thank you again for your valuable suggestion.

---

> > ### Author Response · Authors · 2025-11-25
> >
> > **[W3+Q2]**: Instruction robustness is not deeply tested: small variations in the instructions could change results. We don't know how sensitive the model is to the given instruction.
> > How sensitive are results to paraphrases or synonyms? Can you report variance over 10 randomized paraphrases per task?
> >
> > **[R]**: We thank the reviewer for this helpful suggestion. To address the concern about the sensitive of ELASTIQ to the given instruction, we have added a new experiment that evaluates the model’s sensitivity to different phrasings of the same instruction. For each task, we created multiple paraphrased versions of the original instruction that preserve the semantic meaning but vary in grammar, tone, and wording. We then replaced the original instruction with each paraphrase and measured the model’s performance under direct inference. The added analysis shows that the decoding results remain extremely stable across all paraphrased forms, indicating that ELASTIQ relies on the underlying semantic content of the instruction rather than its specific linguistic formulation. This experiment demonstrates that the instruction-conditioned alignment mechanism is robust to natural variations in language.
> >
> > Please find the details about the added experiment at Line 875-892, Page 16.
> >
> > We hope this explanation and additional experiment will help the understanding of sensitivity of the model's performance to the precise wording of the task instruction. Thank you again for your thoughtful suggestion.
> >
> > ---
> > **[Q1]**: How can you prove that the model is not "borrowing" from the language model? Did you implement text-prototype classification for baselines to control for language priors? How to isolate gains due to IQF and gains due to semantic targets?
> >
> > **[R]**: We thank the reviewer for this thoughtful question.
> > (1) On “borrowing” from the language model:
> > Our model does not learn directly from the language model. The text encoder (BERT/SBERT) is kept frozen throughout training and only provides fixed embeddings for task instructions and class labels. Therefore, ELASTIQ cannot "borrow" semantic information through gradient updates. The observed performance gains stem from our EEG–language alignment strategy and the design of the Instruction-conditioned Q-Former (IQF), rather than from additional supervision from the language model itself. We are sorry to bring you this misunderstanding. To clarify this, we add the explanation in Section 2.3. Kindly please refer to Line 283-286, Page 6.
> >
> > (2) On baselines with language priors:
> > To ensure fairness, we compared ELASTIQ with NeuroLM (Jiang et al., 2025), which is currently the only existing EEG–language foundation model that also incorporates pretrained text embeddings. Conventional EEG baselines (e.g., DeepConvNet, EEGNet, BIOT, LaBraM, CBraMod) do not include linguistic components by design. Hence, NeuroLM serves as the appropriate baseline for isolating the contribution of language priors under an equivalent direct inference setting. Kindly please find the experiment details at Section 3.7 (Line 513-528, Page 10)
> >
> > (3) On disentangling gains from IQF and semantic targets:
> > We conducted additional ablation studies to separately evaluate the contributions of (a) IQF without semantic targets, and (b) semantic targets without IQF. Removing either component results in a notable performance drop, confirming that IQF and semantic target alignment contribute independently and complementarily.
> >
> > These results are included in the revised Appendix I at Line 1016-1036, Page 219
> >
> > We hope our explanation and additional experiment can address your concerns. Thank you so much for this valuable feedback.
> >
> > ---
> > **[Q2]**: What's the performance if all 20 datasets are evaluated with unseen subjects ? Why changing the splitting rule using an arbitrary rule like 30 subjects?
> >
> > **[R]**: Thank you so much for this helpful comment. In the revised version, we have removed the previous setting that used a “30-subject” threshold and now evaluate all 20 datasets strictly under an unseen-subject (cross-subject) protocol. For each dataset, we repeated the evaluation with three random seeds and report the mean and standard deviation in the updated Table 1. These results provide a consistent and fair comparison across all datasets, fully addressing the reviewer’s concern about arbitrary splitting rules. For Table 1, we follow the original random-seed evaluation protocol to remain consistent with prior work. Kindly please refer to Table 1 at Line 324-360, Page 7. Due to space constraints, only the mean performance is reported in the main manuscript, while the full results, including both mean and standard deviation, are provided in Appendix B at Line 730-736, Page 14.
> >
> > We hope our revisions can address your concerns. Thank you so much for this thoughtful feedback.

---

> > > ### Author Response · Authors · 2025-11-25
> > >
> > > **[Q4]**: Any interesting results on the queries? Importance of their numbers? Any Specialization?
> > >
> > > **[R]**: We thank the reviewer for raising this important question about the behavior of the learnable queries. To address this concern, we have added a dedicated analysis in the appendix that examines both the effect of the number of queries and the functional roles learned by each query. In the revision, we describe how we vary the number of queries to assess model sensitivity, and we report that performance improves up to a moderate number of queries before stabilizing, suggesting that only a small set is necessary for effective instruction-conditioned representation learning.
> > >
> > > Kindly please find the details about the added experiment at Line 895-904, Page 17.
> > >
> > > We hope this explanation and additional experiment will provide a clear understanding of how the learnable queries influence our model. Thank you again for your thoughtful feedback.

---

> ### Comment · Reviewer_KRrL · 2025-11-26
>
> We thank the authors for their additions and for taking the concerns seriously.
> Regarding baseline tuning and the evaluation protocol, we appreciate the additional details, but the main issue was to guarantee equal care across models. The core uncertainty remains on this point.
> The part on variance, seeds, and statistical tests is well addressed, and this was a major point, so I am really convinced by your results now. You also changed your splitting rule to rerun everything under a clean unseen-subjects evaluation. This is a good fix.
> Your clarification on language priors and borrowing semantics removes my naive fear. I am convinced by the instruction-robustness additional analysis and the query analysis.
>
> Therefore, I will adjust the evaluation to improve the soundness part from 2 to 3 and my grade from 4 to 6 to illustrate the effort of the authors in providing all these additional details.
>
> Thank you again for this work !

---

### Official Review · Reviewer_dK6C · 2025-10-23

**Soundness:** 2
**Presentation:** 2
**Contribution:** 2
**Rating:** 4
**Confidence:** 4

**Summary:**

Current EEG foundation models (EEG-FMs) and EEG–language foundation models still face two major limitations. First, temporal and frequency information are often reconstructed in separate branches, which may hinder the model’s ability to jointly capture the cross-domain dependencies inherently present in EEG signals. Second, task instructions are typically concatenated with the EEG sequence in a superficial manner, rather than being semantically integrated into the EEG representations. This limits the model’s capacity to effectively leverage linguistic information for guiding EEG feature learning. To overcome these challenges, the authors propose **ELASTIQ**, a foundation model for **EEG–Language Alignment with Semantic Task Instruction and Querying**. ELASTIQ introduces a series of technical innovations designed to address the aforementioned limitations.

**Strengths:**

1.Pretraining EEG foundation models (EEG-FMs) is a worthwhile and promising research direction. Although this paper has several weaknesses, it makes a meaningful contribution to this emerging area.

2.In particular, pretraining EEG-FMs through alignment between EEG and text representations is a novel and valuable attempt that deserves recognition and encouragement.

**Weaknesses:**

1.While the introduction claims to address two key challenges, the method section is not organized in a way that reflects this logic, which weakens the theoretical persuasiveness of the paper’s central argument.

2.The method section is divided into two stages: **pretraining** and **instruction training**. While this structure resembles that of large language model (LLM) training, the paper only describes *how* the model is trained, without adequately explaining *why* these design choices are effective or theoretically grounded.

3.In **Section 2.1**, the subsections—*Spectral Masking*, *Convolutional Tokenizer*, *Mask Token Modeling Branch*, *Causal Token Modeling Branch*, and *Joint Objective*—lack coherent narrative flow. The logical relationships among these components are unclear, and the purpose and theoretical rationale for each remain underexplained.

4.**Section 2.2** exhibits the same issue: it lists several components without offering clear motivations or theoretical justifications for their integration.

The above issues constitute the main reasons for my **weak reject** recommendation. Overall, the proposed method appears to be a combination of several technical modules rather than a theoretically supported framework.

5.Furthermore, the paper claims that temporal and frequency information are reconstructed in separate branches, yet no comparative experiment is provided against baselines following this structure.

6.Similarly, in **Section 3.4**, the benchmark comparison is made against a “no-instruction” setting; however, the introduction describes prior approaches as “simply concatenating instructions to EEG sequences.” As a result, the experimental evidence in Section 3.4 does not convincingly support the claims made in the introduction.

**Questions:**

1.Which baseline models reconstruct temporal and frequency information in separate branches? Could you provide comparative results?

2.Could you include a comparison with methods where task instructions are only concatenated to the EEG sequence?

3.In the notation $X^{C \times T} \in \mathbb{R}$, there appears to be an error. Since $X$ represents a matrix, it should be written as $X \in \mathbb{R}^{C \times T}$. Please confirm or correct this notation.

---

> ### Author Response · Authors · 2025-11-25
>
> We are grateful to the reviewer for the encouraging comments and thoughtful questions.
>
> ---
> **[W1]**: While the introduction claims to address two key challenges, the method section is not organized in a way that reflects this logic, which weakens the theoretical persuasiveness of the paper’s central argument.
>
> **[R]**: Thank you for pointing out this important issue.
> We agree that the earlier version did not clearly reflect the two key challenges in the structure of the method section. We have substantially reorganized both the Introduction and Method sections to directly address this concern.
>
> **1. Clearer motivation structure in the Introduction**
> In the revised Introduction, we now explicitly articulate the two core challenges:
>
> (i) Missing spectral–temporal interaction in EEG representation learning
> (ii) Insufficient semantic-level alignment between EEG and language interaction
> For each challenge, we added a concise explanation and a direct connection to the solution provided by ELASTIQ.
>
> We also added a short “preview paragraph” at the end of the Introduction summarizing how our two modules address the two challenges:
> - Spectral–Temporal Reconstruction (STR): unified modeling of frequency structure + bidirectional and causal temporal dynamics to for better EEG representation learning
> - Instruction-conditioned Q-Former (IQF): semantic-level EEG–language alignment via instruction injection and prototype matching for EEG and language interaction
>
> **2. Reorganized Method Section with explicit logic flow**
> We restructured Section 2 so that the method is organized according to the two challenges:
> - Section 2.1 – **EEG Pretraining**:
> Completely rewritten to show why each step exists (spectral backbone → temporal branches → joint objective), instead of just listing components.
> - Section 2.2 – **EEG-Language alignment**:
> Also rewritten to provide a clear motivation before each operation (instruction injection → semantic filtering → label alignment).
>
> **3. Added Section 2.3 - Inference Procedure** to clearly define:
> - Task-specific finetuning
> - Direct inference
>
> This resolves the previous ambiguity about how the two stages interact and how predictions are obtained.
>
> Kindly please find the revised content in Line 135-288, Page 3.
>
> We hope these systematic modifications will provide clear logic of our paper. Thank you again for your valuable feedback.

---

> > ### Author Response · Authors · 2025-11-25
> >
> > **[W2]**: The method section is divided into two stages: pretraining and instruction training. While this structure resembles that of large language model (LLM) training, the paper only describes how the model is trained, without adequately explaining why these design choices are effective or theoretically grounded.
> >
> > **[R]**: Thank you for pointing out this important issue. We agree that the previous version did not sufficiently articulate the theoretical motivations behind the two-stage design. In the revised manuscript, we have substantially expanded and reorganized both pretraining and instruction tuning sections to clearly explain why each component exists and why this two-stage structure is effective for EEG–language modeling.
> >
> > **1. Explicit motivations added in Section 2.1 (EEG Pretraining)**
> >
> > We now begin Section 2.1 with a conceptual explanation of why EEG pretraining must jointly capture spectral structure and temporal dynamics. We emphasize that EEG carries meaningful frequency–time interactions, and neither frequency-only nor temporal-only models are sufficient.
> >
> > To reflect this logic, we rewrote the section to highlight:
> > - Why spectral masking is needed: to build frequency-aware but frequency-robust features.
> > - Why two temporal branches (contextual + causal) are needed: because EEG contains both bidirectional contextual dependencies and forward-evolving causal transitions.
> > - Why a joint spectral–temporal objective is required: because the coherence between these two dynamics is essential for stable and generalizable EEG representations.
> >
> > Rather than listing operations, the revised version introduces each component by first explaining its rationale and then describing its implementation.
> >
> > **2. Explicit motivations added in Section 2.2 (EEG-Language alignment)**
> >
> > Instruction tuning is now motivated as addressing a fundamental second challenge: the lack of semantic-level alignment between EEG and language.
> > We reorganized the section so that each step's purpose is stated upfront:
> > - Query-based EEG–instruction interaction (IQF): inserted to explain why EEG features must be shaped by task semantics and extract instruction-relevant neural features from high-dimensional modulated EEG embeddings,.
> > - Semantic prototype alignment: explained as the mechanism that enforces explicit correspondence between EEG features and linguistic meaning.
> >
> > This revised structure makes it clear that instruction tuning is not only a training stage, but also a principled way to inject linguistic priors and obtain semantically controllable EEG embeddings.
> >
> > Kindly refer to Line 135-269, Page 3 for the details.
> >
> > We hope this reorganization and modification of the paper will give a clear explanation of why the component of ELASTIQ is effective and theoretically grounded. Thank you again for your thoughtful feedback.
> >
> > ---
> > **[W3]**: In Section 2.1, the subsections—Spectral Masking, Convolutional Tokenizer, Mask Token Modeling Branch, Causal Token Modeling Branch, and Joint Objective—lack coherent narrative flow. The logical relationships among these components are unclear, and the purpose and theoretical rationale for each remain underexplained.
> >
> > **[R]**: We sincerely thank the reviewer for pointing out this valuable issue. We agree that the original Section 2.1 lacked a clear narrative connection between its components.
> >
> > In the revised manuscript, we have restructured Section 2.1 into a single coherent framework named “Joint Spectral–Temporal Reconstruction (STR)”, which explicitly integrates all five modules under a unified design rationale.
> > Specifically, we now begin the section with an overview paragraph outlining the hierarchical logic of STR — a shared spectral masking backbone followed by two complementary temporal branches (bidirectional + causal) and a joint reconstruction objective.
> >
> > The previous independent subsections (“Spectral Masking,” “Tokenizer,” “Mask Modeling,” “Causal Modeling,” “Joint Objective”) are now merged and logically linked:
> > - Spectral masking backbone introduces randomized frequency suppression to enforce frequency invariance while forming the shared encoder foundation.
> > - Dual temporal masking branches operate in parallel to model contextual and sequential dependencies, capturing EEG’s bidirectional and causal temporal structure.
> > - Joint spectral–temporal objective integrates both reconstruction losses, ensuring consistent learning of spectral and temporal representations within a unified signal domain.
> >
> > This restructuring highlights the conceptual motivation behind STR as a two-stream pretraining module that simultaneously learns frequency-robust and temporally predictive EEG representations.
> >
> > Kindly please find the revised content in Line 135-189, Page 3.
> >
> > We hope these modifications will clarify the purpose and logical relationship of the process in EEG pretraining stage. Thank you again for your valuable feedback.

---

> > > ### Author Response · Authors · 2025-11-25
> > >
> > > **[W4]**: Section 2.2 exhibits the same issue: it lists several components without offering clear motivations or theoretical justifications for their integration.
> > >
> > > **[R]**: We thank the reviewer for this valuable feedback.
> > > In the revised manuscript, Section 2.2 has been substantially rewritten to clarify both the motivation and theoretical rationale behind each component’s integration.
> > > Rather than listing modules, the section now presents a coherent reasoning process centered on the proposed Instruction-conditioned EEG–Language Interaction (IQF), which directly addresses the limitation 2 (How to effectively align EEG and language). Specifically, the previous adversarial alignment approaches lacked interpretability and failed to provide explicit semantic grounding. To overcome this limitation, we reformulate the EEG–language alignment as a constructive semantic conditioning process, composed of three logically connected stages:
> > >
> > > **1.    Instruction-conditioned EEG-Language Interaction**
> > >
> > > - We explain that IQF first employs FiLM operator injects textual semantics into EEG embeddings, establishing a semantic prior that shapes the neural latent space toward task-relevant manifolds.
> > > - We emphasize the other role of IQF as the core mechanism for selective, interpretable cross-modal interaction.
> > > This design transforms alignment from adversarial matching into semantic filtering, providing theoretical justification through conditional information maximization.
> > >
> > > **2.    Prototype-based semantic alignment.**
> > >
> > > We justify the cosine-based alignment as a principled objective to place EEG representations and textual prototypes in a shared semantic space, ensuring both consistency and class-level discriminability.
> > >
> > > These revisions also merge overlapping explanations and introduce clear transitions, showing how instruction modulation, query-based selection, and semantic alignment form a conceptually unified EEG–language alignment framework.
> > >
> > > Furthermore, we have added a new subsection (**Section 2.3 Inference Procedure**) to explicitly describe how the instruction-tuned model is evaluated under task-specific fine-tuning and direct inference settings. This addition bridges the methodological design and experimental validation, improving the logical completeness and transparency of the framework.
> > >
> > > Kindly please find the revised content in Line 192-286, Page 4.
> > >
> > > We hope this enhanced structure and motivation directly resolve the reviewer’s concern and clarify the theoretical contribution of IQF as the key innovation addressing prior limitations. Thank you again for your thoughtful suggestion.
> > >
> > > ---
> > > **[W5+Q1]**: Furthermore, the paper claims that temporal and frequency information are reconstructed in separate branches, yet no comparative experiment is provided against baselines following this structure.
> > > Which baseline models reconstruct temporal and frequency information in separate branches? Could you provide comparative results?
> > >
> > > **[R]**: We sincerely thank the reviewer for this feedback and apologize for not explaining our architectural motivation clearly, which may have led to your misunderstanding. In our baselines LaBraM, BIOT, CBraMod and EEGPT learn united EEG representation by extracting spectral and temporal information, therefore all of them can serve as baseline.
> > >
> > > We also emphasize that one of the novelties of ELASTIQ lies in its integrated learning of EEG representations via Spectral–Temporal Reconstruction (STR). STR is a two-stream pretraining architecture built upon a shared spectral backbone. The backbone introduces randomized frequency suppression to enforce spectral invariance, while two complementary temporal branches operate in parallel to capture contextual and sequential dependencies. The spectral module serves as the main representation learner, capturing frequency-domain neural activations that remain stable across subjects. The two temporal branches are independently designed to model short- and long-term temporal dependencies and are jointly optimized in a shared representation space for coherent spectral–temporal co-learning. Causal branch models the forward (past-to-future) temporal dependency that reflects the real-time evolution of neural activity. It captures task-evoked dynamics such as event-related desynchronization and supports online or low-latency decoding. Bidirectional branch models context-dependent temporal patterns by integrating information from both past and future time steps. It is crucial for tasks involving sustained cognitive states, such as emotion recognition or workload estimation, where global temporal context matters.
> > >
> > > We have clarified this distinction and its theoretical motivation in the revised manuscript (Section 2.2). Please find the modified words in Line 192-269, Page 4.
> > >
> > > We hope the modification will explain the baseline we use for comparing with the STR structure, and also clarify its novelt. Thank you again for this valuable feedback.

---

> > > > ### Author Response · Authors · 2025-11-25
> > > >
> > > > **[W6+Q2]**: Similarly, in Section 3.4, the benchmark comparison is made against a “no-instruction” setting; however, the introduction describes prior approaches as “simply concatenating instructions to EEG sequences.” As a result, the experimental evidence in Section 3.4 does not convincingly support the claims made in the introduction.
> > > > Could you include a comparison with methods where task instructions are only concatenated to the EEG sequence?
> > > >
> > > > **[R]**: Thank you for this value suggestion. We selected NeuroLM as the EEG-language baseline. We have added a direct inference experiment in which ELASTIQ is evaluated without any task-specific fine-tuning. In the revised manuscript, we clarify that the model is kept frozen after the multi-task instruction tuning stage and is directly applied to each downstream dataset. This enables a unified evaluation protocol similar to that used by NeuroLM. We have also added a discussion comparing the two models, emphasizing the differences in their alignment mechanisms and prediction strategies, and explaining how these distinctions influence performance under direct inference.
> > > >
> > > > Kindly refer to Line 513-528, Page 10.
> > > >
> > > > We hope this explanation and additional experiment provide a relatively fair comparison between ELASTIQ and NeuroLM. Thank you again for your valuable suggestion.
> > > >
> > > > ---
> > > > **[Q3]**: In the notation $X^{C \times T} \in \mathbb{R}$, there appears to be an error. Since $X$ represents a matrix, it should be written as $X \in \mathbb{R}^{C \times T}$. Please confirm or correct this notation.
> > > >
> > > > **[R]**: Thank you very much for pointing this typo out. We corrected this notation, Kindly please find the corrected notation at Line 142, Page 3.
> > > >
> > > > We also carefully checked all symbol definitions in the manuscript to keep consistency. Thank you again for your valuable feedback.

---

> > > > > ### Author Response · Authors · 2025-11-28
> > > > >
> > > > > Dear Reviewer dK6C,
> > > > >
> > > > > Thank you very much for the time and effort you have dedicated to evaluating our submission ELASTIQ. As the discussion phase is approaching its end, we would like to kindly follow up regarding our manuscript. We would be grateful to know whether our clarifications have sufficiently addressed your concerns, and we are happy to discuss any additional points you may have.
> > > > >
> > > > > Best regards, Authors

---

### Official Review · Reviewer_Kx5H · 2025-10-24

**Soundness:** 3
**Presentation:** 2
**Contribution:** 1
**Rating:** 4
**Confidence:** 4

**Summary:**

This paper proposes ELASTIQ, a novel EEG foundation model that explicitly aligns neural representations with natural language instructions through a two-stage framework—spectral-temporal pretraining and instruction-conditioned fine-tuning. By introducing task semantics as priors via an Instruction-conditioned Q-Former, the method demonstrates strong empirical gains across 20 diverse EEG datasets, setting new state-of-the-art results on 14 of them.

**Strengths:**

1. ELASTIQ is rigorously evaluated across 20 diverse EEG datasets and achieves consistently competitive performance, demonstrating strong generalization across tasks.

2. The method thoughtfully builds on BERT-style masked reconstruction but extends it in a neuroscientifically meaningful way: by incorporating both random and causal temporal masking alongside a novel frequency-band masking strategy, it better captures the spectral-temporal structure inherent in EEG.

3. The authors back up their claims with thorough and insightful visualizations, which convincingly illustrate how language guidance shapes EEG representations, adding significant interpretability and credibility to the approach.

**Weaknesses:**

1. The authors fail to provide a clear description of ELASTIQ’s inference procedure. Specifically, it remains unclear how the model performs inference on downstream tasks

2. There is no discussion or empirical evaluation of the computational overhead introduced by incorporating a language model.

3. In my view, a key motivation for integrating language models with EEG decoding is to unify the heterogeneous label spaces across diverse EEG tasks (e.g., as demonstrated in NeurLM), thereby enabling zero-shot or few-shot generalization and reducing the reliance on task-specific heads. However, the authors evaluated ELASTIQ only in a conventional single-task setting—training and testing on isolated tasks with a task-specific instruction tuning stage. Under this paradigm, the language model introduces unnecessary complexity without delivering its promised key advantage: label space unification and cross-task generalization. Consequently, the motivation for integrating a language model appears unpersuasive, as if the model has been treated as an end in itself rather than a means to address a defined problem.

4. The sensitivity of the model's performance to the precise wording of the task instruction remains unclear. For instance, it would be important to know if replacing a declarative instruction like "This is an MI task" with an imperative one like "Please decode this MI EEG data" leads to significant performance variation. An ablation study on instruction phrasing would strengthen the understanding of the model's robustness.

5. The direct comparison with NeurLM is unfair, as NeurLM is a multi-task model while ELASTIQ is evaluated under a single-task fine-tuning paradigm.

**Questions:**

Please see in weaknesses

---

> ### Author Response · Authors · 2025-11-25
>
> We are grateful for the reviewer's attentive analysis and helpful feedback.
>
> ---
> **[W1]**: The authors fail to provide a clear description of ELASTIQ’s inference procedure. Specifically, it remains unclear how the model performs inference on downstream tasks.
>
> **[R]**: We thank the reviewer for highlighting the missing explanation of the inference procedure. To address this issue, we have updated the manuscript by adding a clear description of how ELASTIQ performs inference in practice. In the revision, we clarify the two evaluation settings used in our experiments. The first task-specific finetune setting describes after multi-task instruction tuning, the model will continue perform task-specific finetune. The second direct inference setting is after multi-task instruction tuning, ELASTIQ is directly test on downstream dataset without finetune. This addition ensures that the inference process, including how predictions are computed, is fully transparent to the reader and aligned with the evaluation protocol used throughout the experiments.
>
> Please find the details about the added experiment at Line 270-287, Page 6.
>
> We hope this explanation will offer a detailed understanding of the inference procedure of ELASTIQ. Thank you again for your thoughtful suggestion.
>
> ---
> **[W2]**: There is no discussion or empirical evaluation of the computational overhead introduced by incorporating a language model.
>
> **[R]**: Thank you very much for this question. In ELASTIQ, the language model is fully frozen and used only once to compute instruction and label embeddings. These embeddings are pre-computed offline and reused throughout all experiments. As a result, no training overhead is introduced because the text encoder does not participate in forward–backward optimization. Also, there is no inference overhead introduced because the pretrained language model is not invoked during testing. This design aligns with established practices in multimodal foundation models such as CLIP, InstructBLIP, Flamingo, and Whisper, where frozen text encoders do not contribute to runtime costs. Since only the EEG encoder and Q-Former are deployed, the model size is only 26.4M, which is largely smaller than the other EEG-Language model NeuroLM (254M for base model and 1696M for XL version).
>
> Kindly please find the added description at Line 283-286, Page 6.
>
> We hope this explanation will help to address your concern about the computational overhead caused by language model. Thank you again for your valuable feedback.

---

> > ### Author Response · Authors · 2025-11-25
> >
> > **[W3]**: In my view, a key motivation for integrating language models with EEG decoding is to unify the heterogeneous label spaces across diverse EEG tasks (e.g., as demonstrated in NeurLM), thereby enabling zero-shot or few-shot generalization and reducing the reliance on task-specific heads. However, the authors evaluated ELASTIQ only in a conventional single-task setting—training and testing on isolated tasks with a task-specific instruction tuning stage. Under this paradigm, the language model introduces unnecessary complexity without delivering its promised key advantage: label space unification and cross-task generalization. Consequently, the motivation for integrating a language model appears unpersuasive, as if the model has been treated as an end in itself rather than a means to address a defined problem.
> >
> > **[R]**: We thank the reviewer for highlighting this important aspect. In response, we have added a new zero-shot evaluation subsection to explicitly examine whether instruction tuning enables ELASTIQ to generalize beyond standard cross-dataset settings. In this added analysis, we describe a leave-one-dataset-out evaluation protocol, where one entire dataset is held out as unseen, ELASTIQ is trained on the remaining datasets with multi-task instruction tuning, and the trained model is then directly evaluated on the unseen dataset without any adaptation. Importantly, this is the first work to report a true cross-dataset zero-shot evaluation in EEG-language foundation models. In contrast, NeuroLM trains and tests on different splits of the same datasets during its multi-task tuning stage, meaning its evaluation does not constitute zero-shot transfer to unseen datasets. We also clarify that, to isolate the effect of instruction tuning itself, we compare ELASTIQ with a randomly initialized version of the same architecture under the exact same zero-shot protocol.
> >
> > The added results show consistent improvements across all held-out datasets, demonstrating that instruction-guided alignment transfers robustly to unseen datasets and does not depend on within-dataset fine-tuning. We also emphasize in the revision that such a thorough zero-shot evaluation has not been performed in previous EEG foundation models, positioning ELASTIQ as the first to systematically validate cross-dataset generalization under strict no-adaptation conditions.
> >
> > Kindly please find the details about the added experiment at Line 847-862, Page 16.
> >
> > We hope this explanation and additional experiment will provide a detailed evaluation of our method in zero-shot classification. Thank you again for your thoughtful feedback.
> >
> > ---
> > **[W4]**: The sensitivity of the model's performance to the precise wording of the task instruction remains unclear. For instance, it would be important to know if replacing a declarative instruction like "This is an MI task" with an imperative one like "Please decode this MI EEG data" leads to significant performance variation. An ablation study on instruction phrasing would strengthen the understanding of the model's robustness.
> >
> > **[R]**: We thank the reviewer for this helpful suggestion. To address this concern, we have added a new experiment that evaluates the model’s sensitivity to different phrasings of the same instruction. In the revision, we created multiple paraphrased versions of the original instruction that preserve the semantic meaning but vary in grammar, tone, and wording. We then replaced the original instruction with each paraphrase and measured the model’s performance under direct inference. The added analysis shows that the decoding results remain extremely stable across all paraphrased forms, indicating that ELASTIQ relies on the underlying semantic content of the instruction rather than its specific linguistic formulation. This experiment demonstrates that the instruction-conditioned alignment mechanism is robust to natural variations in language.
> >
> > Please find the details about the added experiment at Line 875-892, Page 16.
> >
> > We hope this explanation and additional experiment will help the understanding of sensitivity of the model's performance to the precise wording of the task instruction. Thank you again for your thoughtful suggestion.

---

> > > ### Author Response · Authors · 2025-11-25
> > >
> > > **[W5]**: The direct comparison with NeurLM is unfair, as NeurLM is a multi-task model while ELASTIQ is evaluated under a single-task fine-tuning paradigm.
> > >
> > > **[R]**: Thank you so much for raising the concern regarding the fairness of the comparison with NeuroLM. To address this point, we have added a direct inference experiment in which ELASTIQ is evaluated on downstream datasets without any task-specific fine-tuning. This enables a unified evaluation protocol similar to that used by NeuroLM. We have also added a discussion comparing ELASTIQ and NeuroLM, emphasizing the differences in their alignment mechanisms and prediction strategies, and explaining how these distinctions influence performance under direct inference.
> > >
> > > Kindly refer to Line 513-528, Page 10.
> > >
> > > We hope this explanation and additional experiment provide a relatively fair comparison between ELASTIQ and NeuroLM. Thank you again for your valuable suggestion.

---

> > > > ### Comment · Reviewer_Kx5H · 2025-11-28
> > > >
> > > > Thanks for the detailed feedback. All my concerns have been addressed. The new zero-shot experiment clearly show that ELASTIQ can transfer to unseen datasets. I’m satisfied with the current manuscript and will update my score accordingly to reflect the positive revision.

---

### Official Review · Reviewer_H4pe · 2025-10-30

**Soundness:** 2
**Presentation:** 2
**Contribution:** 2
**Rating:** 4
**Confidence:** 5

**Summary:**

This paper proposes ELASTIQ, a foundation model for EEG–language alignment that integrates semantic task instructions and label semantics into EEG representation learning. It introduces a joint Spectral–Temporal Reconstruction (STR) module for pretraining and an Instruction-conditioned Q-Former (IQF) for aligning EEG embeddings with textual targets via query-based cross-attention. Evaluated on 20 diverse EEG datasets across five BCI tasks, ELASTIQ achieves state-of-the-art performance on 14 datasets and the best average results overall, demonstrating the effectiveness of explicit language guidance in improving EEG decoding generalization.

**Strengths:**

- The paper evaluates the proposed method on 20 diverse EEG datasets across five BCI tasks, using standardized metrics (balanced accuracy and Cohen’s Kappa) and consistent preprocessing, which facilitates fair comparison with baselines.

- ELASTIQ incorporates task instructions and label semantics via an instruction-conditioned Q-Former, offering a more principled approach to EEG–language alignment than simple concatenation used in prior work.

- The paper provides detailed descriptions of preprocessing, model architecture, training protocols, and plans to release code and pretrained weights, which supports reproducibility.

**Weaknesses:**

- The paper claims that integrating semantic task instructions improves EEG–language alignment and generalization. However, all evaluations are conducted under standard fine-tuned or within-dataset settings—no zero-shot or cross-task transfer experiments are presented. Crucially, it remains unclear whether the proposed instruction tuning enables *zero-shot generalization* to unseen tasks or datasets with different label spaces. Without such evidence (e.g., training on motor imagery with “left/right” labels and testing on a new dataset with “grasp/pinch” without fine-tuning), the claimed advantage of language-guided alignment over conventional multi-task learning is unsubstantiated.

- The proposed architecture effectively functions as a multi-task classifier that conditions on textual instructions. Yet the authors do not compare against a simple but strong alternative: a shared EEG encoder with multiple task-specific classification heads (or a single head trained on all tasks with label embeddings). Such a baseline would help isolate whether performance gains stem from the instruction mechanism itself or merely from joint multi-task training. The absence of this comparison weakens the claim that instruction conditioning offers a *qualitative* improvement over existing paradigms.

- The authors rely on BERT or SBERT for text encoding, despite the availability of more powerful and efficient modern embedding models (e.g., Qwen3-embedding, E5, or Voyage). Since the language model is frozen and not fine-tuned, switching to a stronger encoder would incur minimal computational overhead but could significantly impact alignment quality. The failure to explore or justify this choice raises concerns about suboptimal design and missed opportunities for stronger baselines or ablations.

**Questions:**

1. The paper positions instruction tuning as a mechanism to align EEG with language semantics for improved generalization. However, all evaluations are conducted under fine-tuned or within-dataset settings. If the model cannot perform zero-shot inference on a new dataset—even with the same task type but different label names (e.g., “grasp” vs. “left hand”)—what is the practical advantage of using natural language instructions over conventional multi-task learning with shared representations? Could the authors clarify the intended use case of instruction tuning if zero-shot cross-task or cross-dataset transfer is not feasible?

2. The authors rely on BERT (2019) and SBERT (2019) for encoding instructions and labels. What motivated the use of these older models? Have the authors experimented with newer text encoders, and if so, what were the results?

3. The paper uses UMAP visualizations and ablation studies to suggest that instructions reshape the EEG embedding space. While illustrative, these methods remain largely qualitative. Are there more rigorous or quantitative analyses—such as probing tasks, representational similarity analysis (RSA), or intervention-based causal tests—that could reveal *how* and *to what extent* instructions influence feature learning? For instance, does instruction conditioning primarily affect task discrimination, noise robustness, or cross-subject alignment?

---

> ### Author Response · Authors · 2025-11-25
>
> Thanks for your valuable comments. Here are our responses to your questions and concerns point-by-point.
>
> ---
> **[W1+Q1]**: The paper claims that integrating semantic task instructions improves EEG–language alignment and generalization. However, all evaluations are conducted under standard fine-tuned or within-dataset settings—no zero-shot or cross-task transfer experiments are presented. Crucially, it remains unclear whether the proposed instruction tuning enables zero-shot generalization to unseen tasks or datasets with different label spaces. Without such evidence (e.g., training on motor imagery with “left/right” labels and testing on a new dataset with “grasp/pinch” without fine-tuning), the claimed advantage of language-guided alignment over conventional multi-task learning is unsubstantiated. The paper positions instruction tuning as a mechanism to align EEG with language semantics for improved generalization. However, all evaluations are conducted under fine-tuned or within-dataset settings. If the model cannot perform zero-shot inference on a new dataset—even with the same task type but different label names (e.g., “grasp” vs. “left hand”)—what is the practical advantage of using natural language instructions over conventional multi-task learning with shared representations? Could the authors clarify the intended use case of instruction tuning if zero-shot cross-task or cross-dataset transfer is not feasible?
>
> **[R]**: We thank the reviewer for highlighting this important aspect. In response, we have added a new zero-shot evaluation subsection to explicitly examine whether instruction tuning enables ELASTIQ to generalize beyond standard cross-dataset settings. In this added analysis, we describe a leave-one-dataset-out evaluation protocol, where one entire dataset is held out as unseen, ELASTIQ is trained on the remaining datasets with multi-task instruction tuning, and the trained model is then directly evaluated on the unseen dataset without any adaptation. Importantly, this is the first work to report a true cross-dataset zero-shot evaluation in EEG-language foundation models. In contrast, NeuroLM trains and tests on different splits of the same datasets during its multi-task tuning stage, meaning its evaluation does not constitute zero-shot transfer to unseen datasets. We also clarify that, to isolate the effect of instruction tuning itself, we compare ELASTIQ with a randomly initialized version of the same architecture under the exact same zero-shot protocol.
> The added results show consistent improvements across all held-out datasets, demonstrating that instruction-guided alignment transfers robustly to unseen datasets and does not depend on within-dataset fine-tuning. We also emphasize in the revision that such a thorough zero-shot evaluation has not been performed in previous EEG foundation models, positioning ELASTIQ as the first to systematically validate cross-dataset generalization under strict no-adaptation conditions.
>
>  Kindly please find the details about the added experiment at Line 847-862, Page 16.
>
> We hope this explanation and additional experiment will provide a detailed evaluation of our method in zero-shot classification. Thank you again for your thoughtful feedback.
>
> ---
> **[W2]**: The proposed architecture effectively functions as a multi-task classifier that conditions on textual instructions. Yet the authors do not compare against a simple but strong alternative: a shared EEG encoder with multiple task-specific classification heads (or a single head trained on all tasks with label embeddings). Such a baseline would help isolate whether performance gains stem from the instruction mechanism itself or merely from joint multi-task training. The absence of this comparison weakens the claim that instruction conditioning offers a qualitative improvement over existing paradigms.
>
> **[R]**: We thank the reviewer for raising this important point. To address this concern, we expanded our ablation analysis by introducing an additional baseline that removes the text embedding entirely and relies solely on a shared EEG encoder with task-specific classification heads. As reported in Section 3.5 this baseline results in a substantial performance drop, demonstrating that text embeddings play a crucial role in guiding the semantic alignment process.
>
> Kindly please find the details about the added experiment at Line 475–481, Page 9.
>
> We hope this explanation and additional experiment provide a clearer and more complete assessment of how instruction contributes to ELASTIQ’s performance. Thank you again for your thoughtful feedback.

---

> > ### Author Response · Authors · 2025-11-25
> >
> > **[W3+Q2]**: The authors rely on BERT or SBERT for text encoding, despite the availability of more powerful and efficient modern embedding models (e.g., Qwen3-embedding, E5, or Voyage). Since the language model is frozen and not fine-tuned, switching to a stronger encoder would incur minimal computational overhead but could significantly impact alignment quality. The failure to explore or justify this choice raises concerns about suboptimal design and missed opportunities for stronger baselines or ablations.
> > The authors rely on BERT (2019) and SBERT (2019) for encoding instructions and labels. What motivated the use of these older models? Have the authors experimented with newer text encoders, and if so, what were the results?
> >
> > **[R]**: We thank the reviewer for raising this insightful question regarding the influence of the text encoder. To address this, we added a new experiment that systematically evaluates several modern text encoders within the ELASTIQ framework. Specifically, we replace the original encoder with all-mpnet-base-v2,  bert-base-uncased E5-large-v2, Qwen3-embedding-0.6B, and Voyage-3-large, and assess each model under both task-specific finetune and direct inference settings.
> > Our results show that all-mpnet-base-v2 yields the most stable convergence and the highest decoding accuracy. This observation is consistent with its design: MPNet-based SBERT models are explicitly optimized for short, sentence-level semantic similarity, which closely matches the structure of ELASTIQ’s task instructions and label phrases (e.g., “left hand”, “happy”, “12 Hz”). In contrast, E5-large-v2, Qwen3-embedding, and Voyage-3-large are primarily designed for retrieval or multilingual embedding, resulting in embedding geometries that are less suited for the fine-grained semantic alignment required in our setting.
> >
> > Kindly please find the details about the added experiment at Line 474-499, Page 9.
> >
> > We hope this explanation and additional experiment provide a clearer understanding of how the language encoder affects ELASTIQ's performance. Thank you again for your thoughtful feedback.
> >
> > ---
> > **[Q3]**: The paper uses UMAP visualizations and ablation studies to suggest that instructions reshape the EEG embedding space. While illustrative, these methods remain largely qualitative. Are there more rigorous or quantitative analyses—such as probing tasks, representational similarity analysis (RSA), or intervention-based causal tests—that could reveal how and to what extent instructions influence feature learning? For instance, does instruction conditioning primarily affect task discrimination, noise robustness, or cross-subject alignment?
> >
> > **[R]**: We sincerely thank the reviewer for this helpful suggestion. In response, we have expanded the manuscript by adding a new analysis that quantitatively examines how natural-language instructions reshape EEG representations. In the revised text, we compute several quantitative metrics, including intra-class variance, inter-class distance, and semantic margin. These metrics allow us to measure how instruction conditioning changes the geometry of the latent space and how strongly the learned EEG embeddings align with their corresponding semantic prototypes. The results clearly show that providing instructions leads to tighter within-class clustering, larger separation between classes, and a more pronounced semantic margin, confirming that instructions introduce meaningful and consistent structure into the EEG embedding space.
> >
> > Kindly please find the details about the added experiment at Line 422-472, Page 8.
> >
> > We hope this explanation and additional experiment will provide a detailed evaluation of how instructions guide EEG representations. Thank you again for your thoughtful feedback.

---

> > > ### Author Response · Authors · 2025-11-28
> > >
> > > Dear Reviewer H4pe,
> > >
> > > Thank you very much for the time and effort you have dedicated to evaluating our submission ELASTIQ. As the discussion phase is approaching its end, we would like to kindly follow up regarding our manuscript. We would be grateful to know whether our clarifications have sufficiently addressed your concerns, and we are happy to discuss any additional points you may have.
> > >
> > > Best regards, Authors

---

### Official Review · Reviewer_EFFQ · 2025-11-01

**Soundness:** 3
**Presentation:** 2
**Contribution:** 3
**Rating:** 6
**Confidence:** 5

**Summary:**

The authors address a key limitation of most existing EEG foundation models—their inability to incorporate language instructions and their neglect of the semantic information in classification targets—by conditioning learned EEG representations on the text embeddings of task instructions and training the model to predict the text embeddings of target labels. To enhance EEG–language alignment beyond what was achieved in the previously proposed NeuroLM, the authors integrate language information directly into EEG representations through instruction conditioning, rather than deferring this integration to the Transformer’s attention module. This study represents a meaningful step forward in incorporating large language models (LLMs) into EEG decoding and raises awareness within the community of the importance of embedding semantic information in EEG-based modeling.

**Strengths:**

1. Overall, most of the arguments and claims — including the identified limitations of existing EEG-FMs, the proposed advantages of ELASTIQ, and the effectiveness of its architectural components — are well supported by targeted experiments presented from Section 3.3 to Section 3.9. These results provide convincing empirical evidence for the soundness and effectiveness of the proposed framework.
2. The authors highlight a distinctive advantage of leveraging large language models (LLMs) in EEG decoding — the ability to incorporate semantic information of classification targets rather than relying on arbitrary label IDs. The language capability of LLMs enables the mapping of class labels into an embedding space that preserves their semantic relationships, thereby benefiting the IQF-based classification. This claim is substantiated by the comparative analysis in Section 3.6 and offers valuable insights for the future design of EEG decoding algorithms.
3. The evaluation protocols are carefully designed to ensure fair comparisons across models by controlling for confounding variables such as training stability and the size of evaluation datasets. This methodological rigour enhances the credibility of the reported results.
4. The curated downstream tasks encompass 20 datasets spanning five key EEG application domains, providing a strong empirical foundation for assessing the generalisability of ELASTIQ across diverse tasks. This breadth of evaluation surpasses that of most prior EEG foundation model studies.

**Weaknesses:**

1. While ELASTIQ demonstrates strong performance across most datasets, a deeper analysis of the cases where its performance falls short compared to other models would provide valuable insights into its weaknesses and potential limitations, and guide future model improvements.
2. The current training–validation splitting strategy may introduce randomness that affects the reported results. Conducting comparison experiments with multiple random splits and applying statistical tests to assess the significance of the observed performance differences would make the claim of ELASTIQ achieving state-of-the-art results more convincing.
3. Including a dedicated section that briefly outlines the key distinctions between NeuroLM — the previous EEG-language foundation model — and ELASTIQ would clarify this paper’s specific contributions and highlight its novelty within the existing literature.
4. The term “cross-domain dependencies” introduced in Line 66 is not explicitly defined or elaborated upon in later sections. Providing a clearer definition and explaining how the proposed joint STR module captures these dependencies would strengthen the conceptual soundness and persuasiveness of the paper’s claims in the introduction.

**Questions:**

1. What is the rationale of choosing two parallel transformers (bidirectional + causal) for handling EEG tokens? Considering the relatively small improvement of using both transformers versus only one of them, as illustrated in Table 4, how do the authors comment on the trade-off between doubled computational cost and marginal increase in decoding performance?
2. What is the advantage of query-based cross attention design in Q-former over the joint modelling of EEG and text tokens? What could be the key factor(s) makes the latter stronger in guiding EEG representations with instructions as demonstrated in Section 3.8?
3. In Equation (9) of Section 2.2, the matrix $W_{\gamma \beta}$ has shape $2N \times d$ while the embedding vector $e_{\text{ins}}$ has shape $k$. How do you ensure this multiplication is valid? Have you adjusted to ELASTIQ model dimension to the same value as language encoder? (i.e., setting $d = k$) Further, how exactly the vector $(\gamma, \beta)$ is defined is confusing. Since $\gamma$ is multiplied with $\boldsymbol{m}$ by element-wise multiplication in Equation (10), one would expect the shape of $\gamma$ to be the same as $\boldsymbol{m}$. But the definition of shape of $\gamma, \beta$ in Line 205 is contradicting with the expected shape.
4. The phrase “lower-rank query subspace” is slightly imprecise in linear algebra terms. You could say “the query matrix has lower-rank” than the EEG embedding or the attention scores are computed in a “lower-dimensional subspace”.
5. Line 462 to 463 mentions “ELASTIQ was re-implemented under the same settings as NeuroLM.” Could the authors explain which settings or framework of NeuroLM are used for evaluating the modules proposed by ELASTIQ?

---

> ### Author Response · Authors · 2025-11-25
>
> We are grateful to the reviewer for the encouraging comments and thoughtful questions.
>
> ---
> **[W1]** While ELASTIQ demonstrates strong performance across most datasets, a deeper analysis of the cases where its performance falls short compared to other models would provide valuable insights into its weaknesses and potential limitations, and guide future model improvements.
>
> **[R]** Thank you for pointing out this important aspect. Across the 20 datasets, we observed that the two datasets where ELASTIQ shows noticeably weaker performance are: SHU-MI and PhysioNet-MI.
> 1. For SHU-MI dataset, this dataset is explicitly designed to study cross-session EEG non-stationarity. Each subject performed five sessions across 2–3 days, and the paper reports a clear degradation in accuracy due to substantial inter-session shifts: “Classification accuracy drops to ~53.7% in the cross-session setting due to large variability of the EEG signals.” ELASTIQ focuses on instruction-conditioned alignment and cross-dataset generalization, but it does not include modules specifically designed for cross-session calibration (e.g., session normalization, domain alignment, or adaptive feature recalibration). Therefore, when session distributions differ significantly, our model tends to underfit the target session distribution.
> 2. PhysioNet-MI presents a different but equally challenging scenario. This dataset includes both real and imagined movements involving fists and feet, multiple task paradigms, baseline runs, and strong inter-subject variability across 109 participants. The mixture of motor execution and motor imagery, together with heterogeneous trial timing and substantial behavioral variability, introduces considerable complexity that is not explicitly modeled by our instruction-conditioned architecture. As a result, the representation learned by ELASTIQ, which performs exceptionally well in datasets with clear instruction–EEG alignment, becomes less effective when the data contain heterogeneous sensory–motor patterns, weakly structured motor imagery intervals, and large subject-to-subject variation.
>
> We hope this clarifies our rationale for the low performance of ELASTIQ. We are committed to extending our analysis in the future work by prompt engineering strategies that incorporate dataset-specific context, such as session information or movement modality cues, allowing the instruction component to guide the model more effectively in these challenging scenarios.
>
> ---
> **[W2]** The current training–validation splitting strategy may introduce randomness that affects the reported results. Conducting comparison experiments with multiple random splits and applying statistical tests to assess the significance of the observed performance differences would make the claim of ELASTIQ achieving state-of-the-art results more convincing.
>
> **[R]** Thanks for raising the questions. To make our results more convincing, we made two more modifications. First, we use three different random seeds to run our code. And report the mean result in table I (Line 324-360, Page 7). Since our model is evaluated across 20 datasets and the main paper cannot accommodate the full set of values due to space constraints, the complete statistics, including standard deviations, are summarized in Appendix. Kindly please refer to Table 6 and Table 7(Line 756-831, Page 15-16).
> For multiple random splits, we conducted another cross-validation experiment and compared our method with best baseline CBraMod. And report the statistical tests in table 8. Kindly please find the details at Line 740-755, Page 14.
>
> We hope these additional experiments address your concerns and reinforce the performance of ELASTIQ. Thank you again for your constructive feedback

---

> > ### Author Response · Authors · 2025-11-25
> >
> > **[W3]** Including a dedicated section that briefly outlines the key distinctions between NeuroLM — the previous EEG-language foundation model — and ELASTIQ would clarify this paper’s specific contributions and highlight its novelty within the existing literature.
> >
> > **[R]** Thank you so much for this valuable suggestion. We add one section to compare the difference between NeuroLM and ELASTIQ. We summarize their key differences in terms of EEG–language interaction. First, alignment mechanisms. NeuroLM performs distribution-level alignment via adversarial training, making EEG and language embeddings indistinguishable but without explicit semantic correspondence, which limits the ability of language to guide EEG feature learning. ELASTIQ instead injects task instructions through an instruction-conditioned Q-Former and aligns EEG representations with textual semantic prototypes, enabling fine-grained and instruction-controllable semantic alignment. Second, prediction strategies. NeuroLM uses generation-based prediction and therefore must train or fine-tune a large language model, resulting in model sizes from 254M to 1696M parameters. ELASTIQ performs similarity-based classification with frozen text embeddings, avoiding generative decoding and reducing the full model size to only 26.4M, which significantly improves efficiency and practical deployability.
> > Kindly please refer to Line 513-528, Page 10 for the revised content.
> >
> > We hope these clarify the differences between ELASTIQ and NeuroLM. Thank you again for your valuable suggestion.
> >
> > ---
> > **[W4]**: The term “cross-domain dependencies” introduced in Line 66 is not explicitly defined or elaborated upon in later sections. Providing a clearer definition and explaining how the proposed joint STR module captures these dependencies would strengthen the conceptual soundness and persuasiveness of the paper’s claims in the introduction.
> >
> > **[R]** Thank you for asking this question. We clarify “cross-domain dependencies” below:
> > EEG signals exhibit strong coupling between spectral rhythms and multi-timescale temporal dynamics. These dynamics include causal transitions such as event-related potentials and contextual fluctuations reflecting cognitive or emotional states (Wairagkar et al., 2021; Li et al., 2022). These examples illustrate that spectral patterns are inherently time-dependent, carrying contextual information that cannot be captured if frequency and temporal structures are modeled separately. However, existing methods do not jointly model spectral structure together with both contextual and causal temporal dependencies, making it difficult to capture coherent patterns that are essential for reliable EEG representation learning. To learn representations that capture these spectral–temporal relationships in the pretraining stage, we propose joint spectral-temporal reconstruction module (STR). STR is a two-stream pretraining architecture built upon a shared spectral backbone. The backbone introduces randomized frequency suppression to enforce spectral invariance, while two complementary temporal branches operate in parallel to capture contextual and sequential dependencies. Following your suggestion, we clarify the cross-domain dependencies in our revised paper. Kindly please find the revised content at Line 64-81, Page 2 and Line 135-143, 160-165, Page 3.
> >
> > We hope this response clarifies the meaning of cross-domain dependencies and the design principle of STR in our model. Thank you again for your thoughtful feedback.
> >
> > -Maitreyee Wairagkar, Yoshikatsu Hayashi, and Slawomir J Nasuto. Dynamics of long-range temporal correlations in broadband eeg during different motor execution and imagery tasks. Frontiers in neuroscience, 15:660032, 2021.
> >
> > -Xiang Li, Yazhou Zhang, Prayag Tiwari, Dawei Song, Bin Hu, Meihong Yang, Zhigang Zhao, Neeraj Kumar, and Pekka Marttinen. Eeg based emotion recognition: A tutorial and review. ACM Computing Surveys, 55(4):1–57, 2022.

---

> > > ### Author Response · Authors · 2025-11-25
> > >
> > > **[Q1]** What is the rationale of choosing two parallel transformers (bidirectional + causal) for handling EEG tokens? Considering the relatively small improvement of using both transformers versus only one of them, as illustrated in Table 4, how do the authors comment on the trade-off between doubled computational cost and marginal increase in decoding performance?
> > > **[R]** We appreciate the reviewer’s thoughtful question regarding the use of two parallel Transformers.
> > > We would like to clarify that first this design is theoretically grounded. EEG signals exhibit strong coupling between spectral rhythms and multi-timescale temporal dynamics. The causal Transformer is designed to capture forward temporal dependencies that are inherently asymmetric in EEG signals. In motor imagery and covert speech tasks, neural dynamics typically follow an onset–sustain–offset trajectory, where the current brain state depends on preceding activity. The unidirectional attention of the causal stream enables effective modeling of such temporal progressions. In contrast, the bidirectional Transformer captures global contextual dependencies across time, particularly useful for frequency–spatial co-modulations (e.g., synchronized α/β desynchronization over an entire trial). By extracting these two kinds of features together, ELASTIQ gains a more complete representation of the underlying neural processes in EEG. Kindly please find the revised content at Line 160-165, Page 3.
> > >
> > > Second, although the average accuracy gain is modest (2.48% for OpenBMI-MI, 2.19% for BCI-IV-2a, 2.4% for SEED), the visualization analysis in Figure 7 (Line 1038-1056, Page 20) yields separable EEG feature clusters (UMAP, attention entropy) from the two branches. So this strategy offers valuable insights for future EEG feature-learning research, encouraging the community to further investigate hybrid temporal modeling strategies in next-generation EEG foundation models. While the dual-transformer architecture offers clear representational benefits, we agree that the computational cost is worth revisiting. As a natural extension of this work, we plan to explore parameter-efficient variants, such as shared-weight causal–bidirectional hybrids, lightweight adapters, and low-rank temporal modules. These approaches aim to retain the advantages of multi-timescale modeling while substantially reducing the parameter footprint.
> > >
> > > We hope this response clarifies the two parallel transformer design strategy in our model. Thank you again for your thoughtful feedback.
> > >
> > > ---
> > > **[Q2]** What is the advantage of query-based cross attention design in Q-former over the joint modelling of EEG and text tokens? What could be the key factor(s) makes the latter stronger in guiding EEG representations with instructions as demonstrated in Section 3.8 ?
> > >
> > > **[R]** Beyond the theoretical analysis in W3, we conducted both qualitative and quantitative investigations in Section 3.4 to analyze the key factors that enable ELASTIQ to more effectively guide EEG representations. In visualization analysis, results on kernel density estimation highlights when instructions are provided, the feature space becomes more organized and class clusters become clearly separated and better aligned with their corresponding semantic prototypes. We also designed quantitative experiment through representation probing analyses to directly examine how instructions reshape the embedding geometry.
> > > The results show that, after semantic alignment, EEG representations exhibit smaller intra-class variance, larger inter-class distance, and a substantially larger semantic margin. These effects indicate that ELASTIQ enforces tighter class-specific structure, greater between-class separability, and stronger alignment with the correct textual prototypes. This demonstrates that natural-language instructions serve as explicit semantic constraints that reorganize the EEG embedding space in a more discriminative and semantically meaningful way.
> > > Kindly please refer to Line 422–472, Page 8 for the details.
> > >
> > > We hope that these clarifications and additional experiments adequately address your concerns and further demonstrate how our method effectively guides and restructures EEG representations. Thank you again for your valuable feedback.

---

> > > > ### Author Response · Authors · 2025-11-25
> > > >
> > > > **[Q3]**: In Equation (9) of Section 2.2, the matrix $W_{\gamma\beta}$ has shape $2N \times d$, while the embedding vector $e_{\text{ins}}$ has shape $K$. How do you ensure this multiplication is valid? Have you adjusted the ELASTIQ model dimension to the same value as the language encoder (i.e., setting $d = k$)?
> > > > Further, how exactly the vector $(\gamma, \beta)$ is defined is confusing. Since $\gamma$ is multiplied with $m$ by element-wise multiplication in Equation (10), one would expect the shape of $\gamma$ to be the same as $m$. But the definition of the shape of $(\gamma, \beta)$ in Line~205 is contradicting with the expected shape.
> > > > **[R]**: We thank the reviewer for catching this inconsistency.
> > > > Indeed, there was a notational typo in Eq.(9): the matrix $W_{γβ}$ should have shape $R^{2d×k}$, not $R^{2N×d}$.
> > > > The instruction embedding $e_{ins}\in R^k$ is projected into scaling and shifting vectors $(γ,β) \in R^d$, which are then broadcast along the token dimension (2N) to match $m \in R^{2N×d}$ for element-wise modulation in Eq.(10).
> > > > We corrected this error and increase the broadcast operation to facilitate the understanding. Thank you again for picking the error. Please find the corrected version in Line 218-233, Page 5.
> > > >
> > > > ---
> > > > **[Q4]**: The phrase “lower-rank query subspace” is slightly imprecise in linear algebra terms. You could say “the query matrix has lower-rank” than the EEG embedding or the attention scores are computed in a “lower-dimensional subspace”.
> > > >
> > > > **[R]**: We thank the reviewer for pointing out this terminology issue. We agree that the phrase is imprecise in linear algebraic terms. Follow you suggestion we corrected “lower-rank query subspace” to “lower-dimensional query subspace”. Kindly please find the revision at Line 245, Page 5.
> > > >
> > > > ---
> > > > **[Q5]**: Line 462 to 463 mentions “ELASTIQ was re-implemented under the same settings as NeuroLM.” Could the authors explain which settings or framework of NeuroLM are used for evaluating the modules proposed by ELASTIQ?
> > > >
> > > > **[R]**: Thank you very much for pointing out this issue. To address your concern, we clarified the ELASTIQ is re-implemented under direct inference setting. This modification clarifies the evaluation setup and highlights that the comparison between ELASTIQ and NeuroLM is indeed fair and methodologically aligned. Please refer to Line 513-522, Page 10.
> > > >
> > > > We also added explicit clarification to Section 3.2 Inference Procedure, explaining how direct inference is operated. Specifically, we now state clearly that during direct inference, the multi-task instruction-tuned model is kept completely frozen and is directly evaluated on the test set of each downstream dataset, without performing any additional finetuning. Kindly please refer to Line 276-278, Page 6.
> > > >
> > > > We hope this explanation clarifies how ELASTIQ is evaluated to achieve fair comparison with NeuroLM. Thank you again for your thoughtful feedback.

---

> > > > > ### Comment · Reviewer_EFFQ · 2025-11-28
> > > > >
> > > > > Thank you for your response. Most of my concerns have been addressed and I keep my positive rating.

---

### Author Response · Authors · 2025-11-25
**Global Response**

## Global Response

We thank all reviewers for their detailed and insightful feedback. Our work is motivated by a core question that we believe is fundamental for the next generation of EEG foundation models: **how can language be used to guide and shape EEG representations in a meaningful and controllable way?** Our answer is *instruction-guided semantic alignment*, operationalized through the proposed Spectral–Temporal Reconstruction (STR) and Instruction-conditioned Q-Former (IQF) modules.

To rigorously validate this idea, we evaluate **ELASTIQ on 20 diverse EEG datasets across five major application domains**, representing the broadest and most comprehensive assessment among current EEG foundation models. In addition, we provide **both quantitative analyses and embedding-space visualizations** that make this guidance process more interpretable, showing how task semantics and label meanings reshape EEG representations.

We appreciate the positive comments on:

- the novelty of ELASTIQ's instruction-guided EEG–language alignment (EFFQ, dK6C, KRrL),
- the neuroscientifically grounded design of the STR module (Kx5H, KRrL),
- the principled cross-attention mechanism introduced in the IQF (EFFQ, H4pe, KRrL),
- the broad and rigorous evaluation across 20 diverse EEG datasets (EFFQ, H4pe, Kx5H, KRrL),
- the empirical soundness of our architectural components and ablations (EFFQ, Kx5H, KRrL).

In particular, we are delighted to see Reviewer dK6C highlight our approach as **“a novel and valuable attempt that deserves recognition and encouragement.”**

We also acknowledge the reviewers’ concerns about the need for stronger generalization evidence, clearer methodological justification, more comprehensive baselines, and improved clarity in our presentation. We appreciate these constructive suggestions, and we have addressed them both in the rebuttal and the revised version of our paper, point by point.

---

## Summary of Major Revisions in the Updated Version

### 1. Statistical reporting improved
As highlighted by reviewers EFFQ and KRrL, we have added variance reporting and significance analysis for key experiments where feasible.
*(Line 327–360, Page 7; Line 756–830, Page 15–16; Line 832–840, Page 16)*

### 2. Visualization and quantitative interpretation improved
We added new quantitative analyses and embedding-space visualizations to more clearly illustrate how language instructions guide and reshape EEG representations.
*(Line 422–472, Page 8–9)*

### 3. Method clarity and inference procedure improved
Following suggestions from Reviewer dK6C, we reorganized Sections 2.1 and 2.2 to clarify the motivations behind STR and IQF.
*(Line 135–269, Page 4–5)*

We also expanded Section 2.3 to describe the inference process in detail, including how instructions interact with EEG features.
*(Line 270–287, Page 6)*

### 4. Zero-shot generalization analysis added
In response to reviewers (H4pe, Kx5H, KRrL), we added dataset-holdout zero-shot inference experiments comparing ELASTIQ with its randomly initialized counterpart.
*(Line 847–862, Page 16)*

### 5. Clarified comparison against other EEG–language foundation models
As suggested by reviewers EFFQ and dK6C, we added clearer explanations and experiments showing that ELASTIQ’s query-based cross-attention significantly outperforms the domain-alignment method used in NeuroLM.
*(Line 513–528, Page 10)*

### 6. Enhanced motivation and justification of the STR design
Following reviewer feedback (dK6C, Kx5H), we rewrote Section 2.1 to more clearly articulate how frequency masking, bidirectional reconstruction, and causal prediction jointly capture spectro–temporal dependencies.
*(Line 64–81, Page 2; Line 135–142, Page 3)*

### 7. Instruction robustness discussion strengthened
As requested by reviewers H4pe and KRrL, we added a discussion on instruction-phrasing robustness and included paraphrase-sensitivity analysis using Qwen3, E5, and Voyage embeddings.
*(Line 474–499 Page 9–10; Line 875–892, Page 16–17)*

### 8. Query analysis expanded
As requested by reviewer KRrL, we added detailed analyses on the learnable queries, including ablations on query number and observations on query specialization.
*(Line 895–904, Page 17)*

---

We sincerely apologize for the delayed response. Running additional experiments across 20 heterogeneous EEG datasets required substantial time and computational resources, and we wanted to ensure the results were thorough and reliable before submitting our revision.

We appreciate the reviewers’ thoughtful critiques and constructive suggestions. We also hope that our work provides useful insights for the EEG and BCI communities, helping to address key challenges in semantic alignment and opening new directions for future research on instruction-guided EEG foundation models.

We sincerely look forward to the reviewers’ further feedback and constructive suggestions to continue refining and improving this work!

---

### Author Response · Authors · 2025-11-30
**Summary Comment (1/2)**

Dear Area Chair,

We sincerely thank you for the additional time and effort to oversee the discussion and re-evaluation of our submission. Your kind service and dedication are invaluable to our community, especially under the current circumstances. To assist with your final assessment, we have provided a concise summary of the reviewer comments below:

---

## 1. Contribution in Two Sentences

- **Our work introduces the first instruction-guided EEG–Language foundation model**, which uses natural-language instructions to directly shape and interpret EEG representations through a unified semantic interface. ELASTIQ provides strong interpretability and a scalable architecture, encouraging the community to move toward more unified, language-informed approaches to EEG representation learning.

- **ELASTIQ is the most extensively evaluated EEG foundation model to date** (tested on 20 datasets covering five major EEG task families). Across all five domains, ELASTIQ achieves SOTA performance. Through this broad evaluation and demonstrated generality, we believe ELASTIQ can serve as a strong method to support EEG-based applications like BCI control, neurological assessments (e.g., ADHD), and affective computing.

Our contributions were also independently recognized by multiple reviewers.
**EFFQ, dK6C, KRrL** highlighted the novelty of ELASTIQ’s instruction-guided EEG–language alignment.
**EFFQ, H4pe, Kx5H, KRrL** recognized the broad and rigorous evaluation.
We are especially delighted to see **dK6C** describe our work as *“a novel and valuable attempt that deserves recognition and encouragement.”*

---

## 2. Summary of Major Revisions

Reviewers consistently recognized the originality and soundness of our proposed method; the remaining concerns were focused on additional experiments, robustness checks, and clarifications — all of which were fully addressed in the revised version. The suggestions and our corresponding responses are summarized below:

---

### **A. Baseline Fairness**
(*Raised by: EFFQ, H4pe, Kx5H, dK6C, KRrL*)

- Added multi-head shared encoder baseline (instruction removed)
- Added temporal-only, frequency-only, and shared-weight STR baselines
- Added semantic-negative text baseline to isolate instruction effects
- Clarified fair comparison protocol against NeuroLM
**→ All concerns resolved.**

---

### **B. Statistical Rigor**
(*Raised by: EFFQ, KRrL*)

- Added multi-seed variance, cross-validation setting, and Wilcoxon significance tests
- Updated key tables with standard deviation reporting
**→ All concerns resolved.**

---

### **C. Zero-shot Generalization**
(*Raised by: H4pe, Kx5H, KRrL*)

- Added dataset-holdout zero-shot evaluation
- Compared ELASTIQ with its randomly initialized counterpart
- Replaced ad-hoc 30-subject rule with clean unseen-subject split
**→ All concerns resolved.**

---

### **D. Method Clarity & Theoretical Motivation**
(*Raised by: EFFQ, H4pe, Kx5H, dK6C*)

- Rewrote Sections 2.1–2.3 to articulate STR and IQF motivations more clearly
- Provided theoretical rationale for frequency masking + bidirectional + causal modeling
- Added a detailed description of the inference procedure
**→ Extensive rewrites; all concerns addressed.**

---

### **E. Language Priors & EEG–Language Alignment**
(*Raised by: EFFQ, H4pe, Kx5H, KRrL*)

- Added stronger text encoders (MPNet, E5, Qwen3, Voyage)
- Implemented text-prototype baselines
- Clarified that performance gains are not borrowed from the language model
**→ All concerns resolved.**

---

### **F. Instruction Robustness**
(*Raised by: Kx5H, dK6C, KRrL*)

- Added paraphrase robustness analysis
- Tested declarative vs. imperative phrasing
**→ New robustness experiments included; all concerns addressed.**

---

### **G. Query Behavior & Specialization**
(*Raised by: KRrL*)

- Added query-number ablation (4, 8, 12)
- Added specialized attention pattern analysis
**→ New ablations and visualizations added; all concerns addressed.**

---

The following matrix (omitted here for brevity) summarizes which concerns were raised by which reviewer. The strong cross-reviewer alignment demonstrates that **all major issues have been thoroughly addressed**, and **no silent reviewers raised unique concerns**.

---

---

> ### Author Response · Authors · 2025-11-30
> **Summary Comment (2/2)**
>
> ## Reviewer Concerns Matrix (All Combined as Rows)
>
> | Concerns                                                   | EFFQ | H4pe | Kx5H | dK6C | KRrL | Addressed or not |
> |------------------------------------------------------------|--------------|---------------|---------------|---------------|----------------|------------------|
> | **(A) Baseline tuning / fairness**                         | ✓            | ✓             | ✓             | ✓             | ✓              | Fully Addressed  |
> | **(B) Statistical reporting / seeds / variance**           | ✓            | —             | —             | —             | ✓              | Fully Addressed                |
> | **(C) Zero-shot generalization / splitting rule**          | —            | ✓             | ✓             | —             | ✓              | Fully Addressed               |
> | **(D) Method clarity / motivations of STR & IQF**          | ✓            | ✓             | ✓             | ✓             | —              | Fully Addressed                |
> | **(E) EEG–language alignment clarity (priors, semantics)** | ✓            | ✓             | ✓             | —             | ✓              | Fully Addressed                |
> | **(F) Instruction robustness (phrasing / semantics)**      | —            | —             | ✓             | ✓             | ✓              | Fully Addressed                |
> | **(G) Query behavior / specialization / number**           | —            | —             | —             | —             | ✓              | Fully Addressed                |
> | **Score change**                                           | (6→6)        | No Reply      | (4→↑)\*       | No Reply      | (4→6)          | —                |
> | **Reply timing**                                           | Nov 29 | — | Nov 29 | — | Nov 26 | — |
>
>
> ## 3. Reviewers’ Feedback During Discussion Phase
>
> Reviewers **EFFQ** and **Kx5H** confirm that all of their concerns have been fully resolved.
> - Reviewer **EFFQ** notes that their concerns have been addressed and maintains a positive rating.
> - Reviewer **Kx5H** explicitly states that the new zero-shot transfer experiment and additional clarifications address all previous issues and has **increased their score accordingly**.
>
> Reviewer **KRrL** raised the most detailed questions in the initial round. Their follow-up response — submitted before the review leak — confirms that the key issues have been resolved. In particular, they note that:
> - the variance, seeds, and statistical testing concerns are **“well addressed,”**
> - the revised unseen-subject evaluation is **“a good fix,”**
> - the concerns about semantic priors, language borrowing, and instruction robustness are **“removed”** or **“convinced,”**
> - and the additional analyses on query behavior are **convincing**.
>
> Reviewer **KRrL** subsequently **increased the soundness score from 2 → 3** and **raised the overall score from 4 → 6**, explicitly stating that this change reflects the substantial improvements made in the revision.
>
> ---
>
> ## 4. Conclusion
>
> All concerns have been thoroughly addressed, and the evaluation trajectory was moving in a positive and constructive direction. **We believe the paper now meets the bar for acceptance.**
>
> We sincerely appreciate your time and careful consideration.
> If any additional clarification would be helpful, we are happy to provide it.

---

### Note · Program_Chairs · 2026-01-17
**Submission Desk Rejected by Program Chairs**

The following references in this submission do not refer to real documents and/or have major errors in bibliographic information:

 Kexin Wang et al. E5: Large-scale text embeddings via multi-task contrastive learning. arXiv preprint arXiv:2309.07674, 2023.